# Colloidal synthesis of large near-bulk InAs quantum dots through seeded and seedless growth using cluster precursors

Ekaterina Salikhova [1,2,3] ✉, Alf Mews[2], Hendrik Schlicke[1,4] & Jan Steffen Niehaus [1] ✉

Quantum dots are high-potential materials for a wide range of applications, particularly in the infrared spectral range, including telecommunications, security, sensing, photovoltaics, and bioimaging. The size and chemical composition of semiconductor quantum dots determine their optoelectronic properties. While II-VI and IV-VI quantum dots are well-studied, the colloidal synthesis of infrared-active III-V quantum dots such as InAs, which are compliant with the European Union directive for restriction of hazardous substances, remains underdeveloped. In particular, the synthesis of larger InAs quantum dots is challenging due to the strong covalent character of In−As bonding, limited control over precursor reactivity, and complex growth pathways. Here we show the synthesis of large, near-bulk InAs quantum dots using atomic clusters as precursors. All nanostructures are synthesized from 'green' commercially available precursors. These results extend the accessible size regime of colloidal InAs nanoparticles to diameters approaching 40 nm and establish a platform for their use in infrared technologies.

Colloidal quantum dots (QDs) form a class of nanomaterials with promising optoelectronic properties, which can cover a wide spectral range reaching from the visible (VIS) to the infrared (IR). The near-infrared (NIR) and the short-infrared (SWIR) regions have attracted growing interest due to a rapidly increasing number of applications in sensing, imaging, and spectroscopy[1]. These technologies are relevant for industrial and consumer sectors, including automotive systems, smartphones, environmental monitoring, material sorting, and enhanced night vision[1–3].

Despite the availability of low-cost silicon-based detectors, their spectral sensitivity is limited to below ≈1100 nm[4,5]. To access longer IR wavelengths, current technologies commonly rely on epitaxially grown materials such as indium gallium arsenide (InGaAs), which are costly due to energy-intensive deposition processes[5,6]. In contrast, IR-active colloidal QDs could offer advantages due to their low-cost wet-chemical synthesis and deposition possibilities[1,2,7].

For applications in consumer electronics and biomedicine, the use of the European Union directive for restriction of hazardous substances (RoHS) compliant QDs is essential[8,9]. A suitable candidate is indium arsenide (InAs), which combines a small bulk bandgap of 0.36 eV (≈ 3444 nm)[10,11] with high electron mobility[12] and a cubic zinc blende crystal structure, which facilitates epitaxial growth of colloidal core/shell heterostructures such as indium arsenide/indium phosphide/zinc selenide/zinc sulfide (InAs/InP/ZnSe/ZnS) QDs interesting for light-emitting technologies[9,13–20]. These properties make InAs QDs an attractive material for advanced optoelectronic applications.

While II-VI and IV-VI QD systems such as mercury telluride (HgTe) and lead selenide (PbSe) have been thoroughly investigated[3,21,22], the colloidal synthesis of III-V QDs like InAs is less developed[20,23–25]. The reason is the higher covalent bonding character, which leads to challenges in precursor reactivity and in the growth control. Hence, harsher reaction conditions like higher temperatures are required,

[1]Fraunhofer IAP-CAN, Department Quantum Materials, Hamburg, Germany. [2]University of Hamburg, Physical Chemistry Department, Hamburg, Germany. [3]Present address: Ghent University, Ghent, Belgium. [4]Present address: Leibniz Institute of Polymer Research Dresden, Dresden, Germany. ✉e-mail: ekaterina.salikhova@ugent.be; jan.steffen.niehaus@iap.fraunhofer.de

challenging size control and uniformity[26–28] Although large InAs QDs grown epitaxially have been reported in 1999[29], the colloidal synthesis of QDs larger than 7 nm, with excitonic absorption peak above 1400 nm, remains a considerable challenge[11,24,30]. Larger sizes are essential for accessing deeper IR wavelengths and approaching bulk-like properties, but require synthetic strategies that can overcome the limitations of precursor chemistry and growth dynamics.

Among III-V materials, InAs has the lowest covalent bonding character, making the synthesis more controllable[28]. However, the most developed synthesis routes for colloidal InAs QDs rely on pyrophoric, toxic, and not readily available arsenic precursors like tris(trimethylsilyl)arsine (TMSAs)[10,11,20]. In 2016, the commercially available and less toxic tris(dimethylamino)arsine (TDMAAs, or amino-As) was identified as a viable alternative[10,31,32]. Although its first use in the InAs QD synthesis was reported in 2000, the initial thermolysis method had several disadvantages compared to TMSAs-based methods, including long reaction times of up to 6 days and the production limited to amorphous nanoparticles (NPs) with sizes up to 2 nm[10,33].

More recently, several synthesis approaches for InAs QDs using the 'greener' amino-As have been developed[13,17,20,34,35]. Commonly, these methods involve InCl_3 in combination with an external reductant, such as diisobutylaluminium hydride (DIBAL-H) or dimethylethylamine aluminium hydride (DMEA-AlH_3)[13,17,20,34]. For example, this was demonstrated when the largest non-elongated InAs QDs reported to date were produced by this method, achieving an absorption peak in the range up to ≈1850 nm and QD sizes up to ≈13 nm[34]. Also, the largest elongated colloidal InAs QDs reported so far (first absorption peak ≈1650 nm) were synthesized using InCl_3, but in this case, with an alternative arsenic precursor[30].

Another promising route for 'green' colloidal InAs QD synthesis is based on indium(I) chloride (InCl). This precursor serves as both an indium source and a mild reducing agent. In solution, it tends to disproportionate into indium(III) chloride (InCl_3) and metallic indium (In)[26]. This eliminates the need for an additional reducing agent, as required in the case of InCl_3[32]. Furthermore, it prevents potential contamination of InAs QDs with metallic cations from the reducing agent. However, the indium side product may reduce the chemical yield, and its temperature-dependent equilibrium is not fully investigated. The synthesis protocol for InAs QDs based on this chemistry was

published by Ginterseder et al., who reported a synthesis route for InAs QDs with a first absorption peak up to 1363 nm[26]. This method was later adapted for the synthesis of InSb and InP QDs[36–38], but has not been further developed for InAs QDs.

A further synthesis strategy is the seeded growth approach, in which small, preformed nanocrystals serve as seeds. In this context, InAs clusters have emerged as promising precursors for controlled overgrowth[13,39,40]. The first demonstration by Tamang et al. in 2016 involved a combination of indium(III) oleate and TMSAs to form such clusters and seeds[39]. Two years later, Srivastava et al. reported a method that relied on InCl_3 and amino-As[13]. In 2021, Kim et al. presented a process based on indium(III) acetate and TMSAs[40]. The largest InAs QDs obtained by the seeded growth method were ≈9 nm[40].

In this work, rather than exploring alternative precursors to grow large InAs QDs, we present a strategy that combines the established InCl precursor chemistry[26] with the use of clusters as readily available precursors. First, we synthesize InAs clusters from "green" InCl and amino-As precursors (Fig. 1a). We then use these clusters to produce small InAs QDs with a low dispersity in size via a heat-up method (Fig. 1b). To grow large QDs, we employ cluster precursors in either a seeded growth (Fig. 1c) or a seedless growth approach (Fig. 1d), and provide insights into the formation mechanism of the QDs from cluster precursors. Both seeded and seedless methods involve alternating cluster injection and annealing steps to drive the growth of non-elongated, non-agglomerated InAs NPs with tunable sizes up to 40 nm, exhibiting optical activity extending beyond ≈2600 nm. The resulting NPs show high crystallinity consistent with the bulk cubic structure of InAs and reach bulk values. Compared to non-elongated InAs QDs[34], our NPs exhibit a further redshift in absorption of ≈1000 nm. The strategy also offers a potential for scalability and integration into continuous flow reactor systems, enabling the industrial production of RoHS-compliant, IR-active QDs. Our findings provide a basis for fundamental studies and applications in optoelectronic technologies, especially in the SWIR region.

## Results and discussion
### Synthesis of InAs cluster precursors
In the past, it has been shown that clusters have already served as precursors for the growth of different QD systems[13,39,41–44]. Also in our

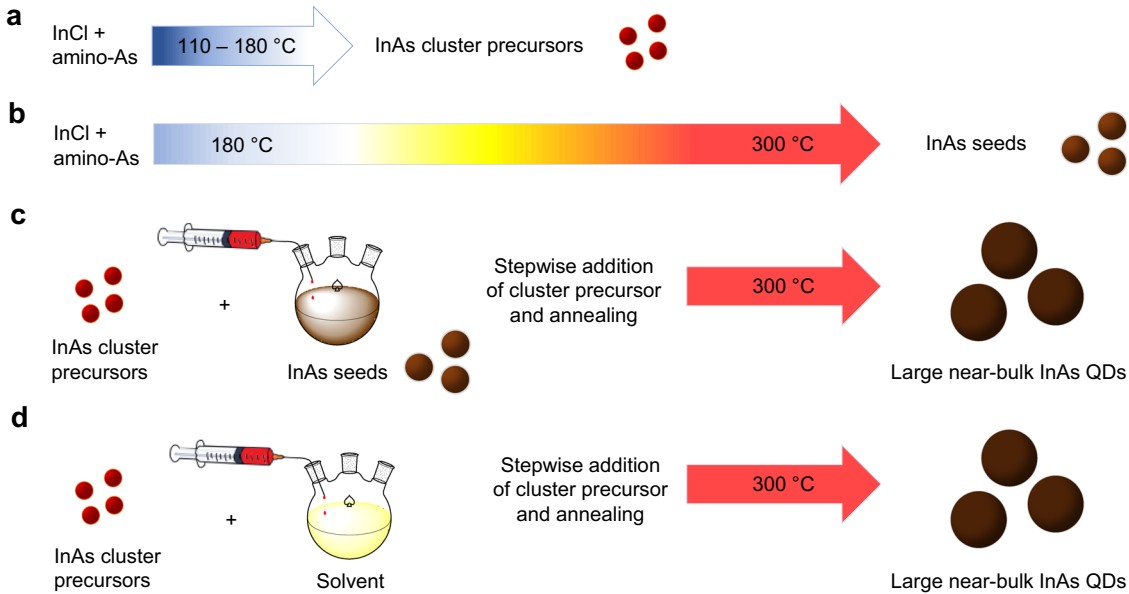

**Fig. 1 | Overview of the different colloidal synthesis methods developed in this work, from InAs cluster precursors (red) to near-bulk InAs QDs (dark brown)[41,42]. a** Synthesis of InAs cluster precursors via hot injection at reaction temperatures ranging from 110 to 180 °C. **b** Heat-up approach for growing small InAs seeds from InAs clusters. **c** Seeded growth and **d** seedless growth methods for synthesizing large near-bulk InAs QDs through multiple injection-annealing cycles.

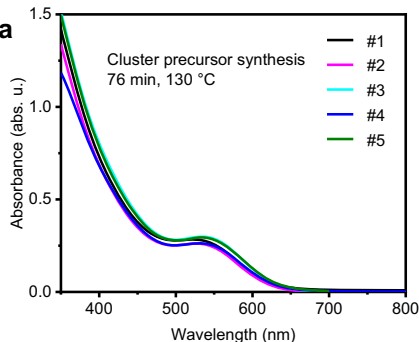
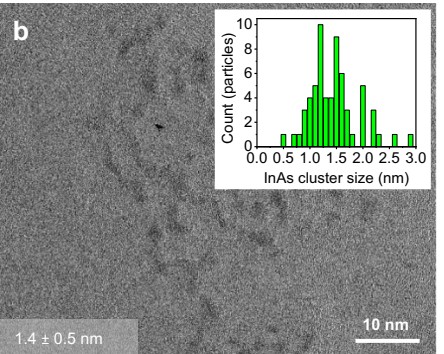
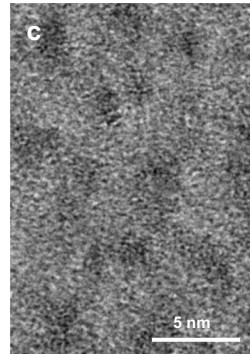

**Fig. 2 | InAs clusters synthesized at 130 °C for 76 min. a** Non-normalized absorption spectra of five InAs cluster precursor solutions (in black, magenta, light blue, blue, and dark green), obtained from five independently repeated experiments, each after removing insoluble byproducts via centrifugation, without further purification[42,50]. **b, c** HR-TEM images of purified InAs cluster precursors, including a size distribution histogram. Samples were synthesized independently five times, and TEM images from one representative batch are shown. Source data are provided as a Source Data file.

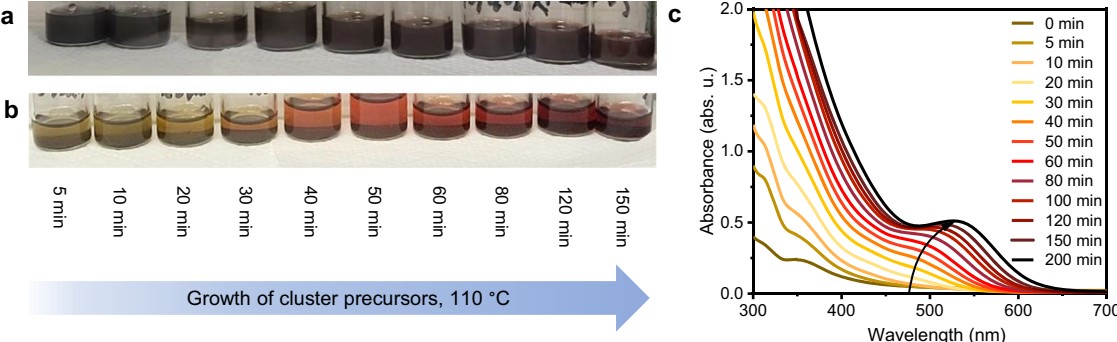

**Fig. 3 | Cluster synthesis at 110 °C[41,42]. a** Photograph of selected aliquots taken directly out of the reaction flask during the synthesis, and **b** after sedimentation. **c** Evolution of absorption properties of aliquot supernatants over time during synthesis, with the arrow indicating the evolution of absorption maxima. Data were not normalized. Measurements were performed after separation of insoluble byproducts via centrifugation, without further purification or dilution. The 0 min aliquot was collected in a separate experiment. Source data are provided as a Source Data file.

approach, the first stage of growing large InAs QDs involved the synthesis of small InAs clusters (Fig. 1a), which exhibited optical activity in the VIS range.

For cluster synthesis, amino-As solution in oleylamine (OAm) was hot-injected into a mixture of InCl in OAm and trioctylphosphine (TOP) at 130 °C, and the reaction was maintained for 76 min. A 3.1:1 ratio of In:As was used, as described in ref. 26 (for reaction yield, see "Methods"). Figure 2a shows absorption spectra of five InAs cluster precursors synthesized using this method, measured after removing insoluble byproducts. The spectra showed minimal variability, demonstrating the method's reproducibility. In the high-resolution transmission electron microscopy (HR-TEM) images (Fig. 2b, c), small nanostructures can be seen. Some of the structures seem to form elongated aggregates (with short and long axes measuring ≈1 and ≈4 nm, respectively), possibly during the deposition process on the grids. Due to the low contrast, neither the crystallinity nor the shape of the clusters can be clearly determined from the HR-TEM images.

A representative selected area diffraction (SAED) pattern of InAs clusters (Supplementary Fig. 1) showed broad, diffuse diffraction pattern, suggesting either a short coherence length or disorder within the clusters' crystal structure.

To investigate the nature of the insoluble byproducts formed during cluster growth, we performed X-ray diffraction (XRD) analysis (see Supplementary Fig. 2). The precipitate primarily contained metallic In, likely arising from the disproportionation of InCl, as reported in ref. 26. Traces of InAs and indium oxide ($In_2O_3$) were also

detected. The formation of $In_2O_3$ could be attributed to side reactions with oxygen-containing impurities in OAm. No pure InCl was detected.

In subsequent experiments, clusters were also synthesized via hot injection at temperatures ranging from 110 to 180 °C. Supplementary Table 1 lists the spectral positions of the first absorption maxima for the cluster samples, measured after removal of insoluble byproducts. A higher reaction temperature and longer reaction time resulted in a redshift, indicating the formation of larger clusters.

To investigate cluster growth more thoroughly, a representative reaction was conducted at 110 °C and monitored by taking aliquots. Figure 3 shows photographs of aliquots taken before (Fig. 3a) and after (Fig. 3b) the sedimentation of insoluble byproducts. The color of the supernatant, ranging from yellow to dark red, indicated the formation of small InAs nanostructures over time. The absorption spectra (Fig. 3c) revealed an evolving absorption band in the VIS range (indicated by an arrow), which became more pronounced and redshifted as the reaction progressed, suggesting a reduction in quantum confinement during cluster growth. Additional absorption features were observed in the ultraviolet (UV) range at ≈315 and ≈350 nm. Prolonging the reaction time beyond 30 min led to a decrease in the prominence of the UV features.

We attribute the absorption features in the UV to molecular structures such as organometallic intermediates or small clusters of a few atoms. In a separate experiment (Supplementary Fig. 3), similar features were observed (Supplementary Fig. 4a). Comparing the corresponding absorption spectrum with those of early-stage aliquots taken during synthesis at 110 °C (see Fig. 3), we observed similar

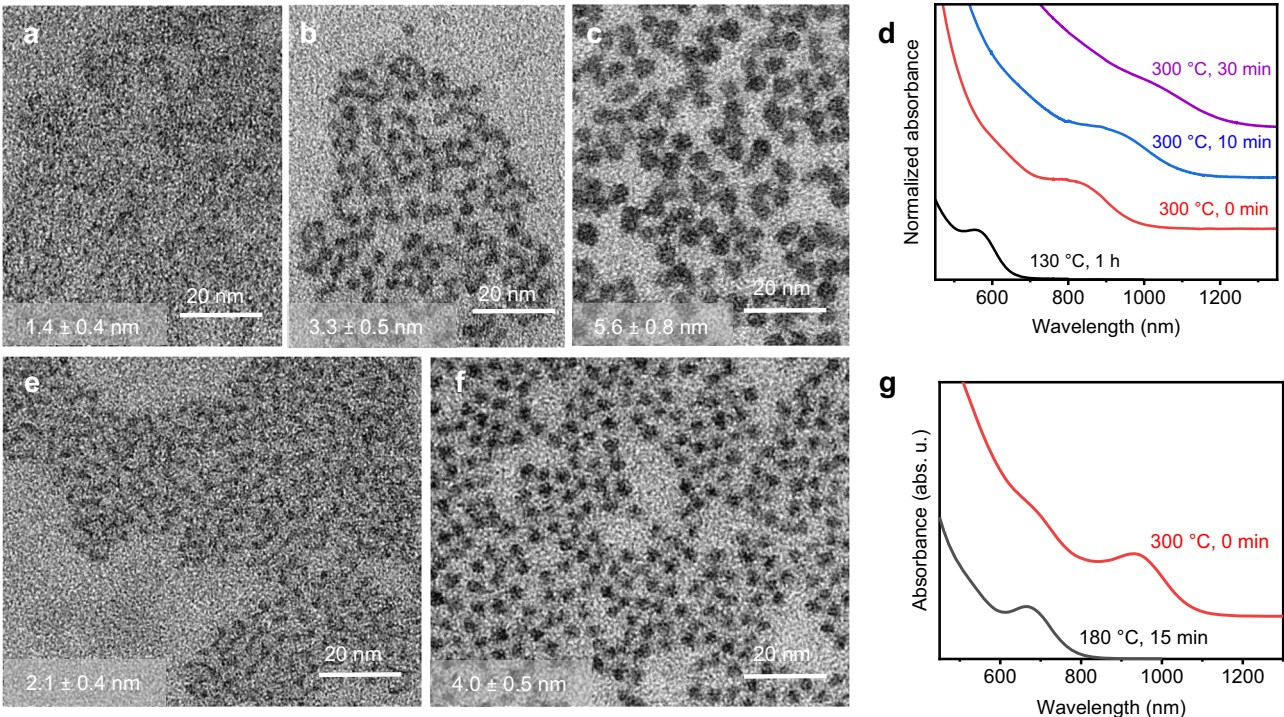

**Fig. 4 | TEM images and corresponding absorption spectra of InAs NPs synthesized using two different heat-up approaches; insoluble byproducts were removed prior to analysis**[41,42]**. a** TEM image of clusters synthesized at 130 °C for 1 h. **b** TEM image of InAs QDs formed by heating the clusters synthesized at 130 °C to 300 °C. **c** TEM image of InAs QDs formed by heating the reaction mixture at 300 °C for an additional 30 min. **d** Absorption spectra of the corresponding NPs (clusters in black and seeds at different reaction times in red, blue, and violet), displayed with an offset and normalized to their respective absorption maxima. **e** TEM image of InAs clusters synthesized at 180 °C for 15 min. **f** TEM image of InAs QDs prepared by heating the clusters synthesized at 180–300 °C. **g** Non-normalized absorption spectra of the corresponding InAs NPs (clusters in black and seeds in red), displayed with an offset. **a**–**c** show samples synthesized once, while **e**, **f** show TEM images from representative batches of four independently repeated heat-up procedures. Source data are provided as a Source Data file.

absorption maxima at ≈315 and ≈350 nm, with a higher intensity ratio (Supplementary Fig. 4b). This suggests that the absorption features correspond to the formation of specific small structures active in the VIS range.

While the formation process of the clusters including the origin of the structures absorbing in the UV is beyond the scope of this work, we demonstrated that the absorption features, and thus the cluster size, could be tuned via adjusting the growth temperature and reaction time. Furthermore, these clusters were found to be chemically stable and could be stored for years at room temperature (RT) within a nitrogen-filled glove box, even after drying and redispersion in the solvent. For example, the absorption data of cluster samples after years of storage are shown in Supplementary Fig. 5. Interestingly, a room-temperature reaction proved unsuitable for producing InAs clusters (see Supplementary Note 1 and Supplementary Fig. 6). The absorption spectra of the particles formed were consistent with metallic nanostructures as presented in Supplementary Figs. 7 and 8.

Nevertheless, the clusters synthesized at 130 °C for 76 min were used as precursors for the growth of larger InAs nanocrystals via seeded growth (Fig. 1c) and seedless growth (Fig. 1d) methods, as will be described in the following.

## Synthesis of InAs seeds

In the second stage of growing large InAs QDs via the seeded growth approach (Fig. 1c), we synthesized small InAs QDs to be used as seeds. These were prepared from clusters in a heat-up process (Fig. 1b). A similar heat-up method was employed by Srivastava et al.[13], heating a so-called "amorphous prenucleation cluster solution", obtained at RT, to a desired temperature between 180 and 290 °C to grow crystalline InAs NPs.

In our study, we explored various combinations of temperatures and reaction times. We first synthesized cluster precursors at either 130 or 180 °C, and then heated a cluster mixture to 300 °C, where the reaction was quenched either immediately or after a defined reaction time by cooling to RT (Fig. 4).

Clusters synthesized at 130 °C for 1 h are shown in Fig. 4a, with additional HR-TEM and high-angle annular dark-field scanning TEM (HAADF-STEM) images provided in Supplementary Fig. 9. Upon subsequent heating to 300 °C, small InAs QDs were formed, as seen in Fig. 4b. The absorption spectra of the particle solution exhibited a pronounced excitonic absorption peak (Fig. 4d, red line). However, a tendency to form 'necks' between the QDs was observed in the TEM images. This is likely due to a small fraction of particles undergoing agglomeration, the deposition process during TEM sample preparation, or specific imaging conditions. Further annealing of the reaction mixture at 300 °C for 30 min led to additional growth of the QDs, but also to a broadening of the size distribution (Fig. 4c). The trend was consistent with the changes in the corresponding absorption spectra (Fig. 4d, blue and violet lines), which showed a redshift and broadening of the absorption onset during longer reaction times at 300 °C.

A significant increase in size and narrowing of size distribution were achieved by initially synthesizing clusters at a higher temperature of 180 °C for 15 min (Fig. 4e), followed by a temperature increase to 300 °C (Fig. 4f). This modification resulted in a redshift of the first absorption peak by ≈150 nm compared to the previous synthesis. The seeds showed lower size dispersity (12 % vs. 15%), and the corresponding XRD and HR-TEM data (Supplementary Figs. 10 and 11) confirmed the crystalline nature of these particles.

Further increasing the final reaction temperature to 320 °C produced larger QDs, but with a broader size distribution and a less pronounced absorption maximum (see Supplementary Fig. 12, double

lines). Therefore, an initial reaction temperature of 180 °C, followed by a final reaction temperature of 300 °C (Fig. 4e, f), was adopted as the standard heat-up approach for seed production (for reaction yield, see "Methods").

We analyzed the insoluble byproducts from the standard heat-up approach using TEM and XRD. From the TEM images (Supplementary Fig. 13), a mixture of size-disperse particles with undefined shapes, along with smaller spherical particles and darker particles, was observed. As in the case of the cluster synthesis, InAs, along with a small amount of $In_2O_3$ and metallic In, were identified from XRD (Supplementary Fig. 14).

Although the reaction mechanism of the heat-up synthesis was not investigated in detail, the method showed high reproducibility, as demonstrated by consistent absorption spectra of four seed solutions prepared via the standard heat-up method, and measured after separation of insoluble byproducts (Supplementary Fig. 15). Elemental analysis of the isolated QDs gave an In:As ratio of 1.78:1 (Supplementary Table 2). The In-rich QDs were in agreement with InAs QDs synthesized from amino-As[13].

In all heat-up experiments described, insoluble byproducts from the initial cluster precursor synthesis were not removed prior to heating. In contrast, as a control, when insoluble byproducts of the cluster synthesis were removed by centrifugation prior to heating, smaller QDs with a broader size distribution and reduced concentration were formed (see Supplementary Fig. 12, dashed lines). TEM revealed the presence of larger, darker particles, likely metallic In (Supplementary Fig. 16), suggesting incomplete reaction in the initial cluster growth. These findings suggest that the byproducts may serve as a reservoir during the heat-up process, influencing the local chemical environment and promoting more uniform nucleation and growth of the QDs. However, prior to using the seeds for the growth of larger InAs QDs in the seeded growth approach, insoluble byproducts were consistently removed by centrifugation.

## Seeded growth synthesis of large InAs QDs

In the last stage of growing large InAs QDs via the seeded growth approach, we used InAs cluster precursors and seed solutions after removing insoluble byproducts. Cluster precursor solutions were continuously injected into a solution of InAs seeds at a constant reaction temperature of 300 °C using a syringe pump. Key parameters, such as cluster and seed size, concentration, and injection rate of clusters, were critical for obtaining non-agglomerated InAs QDs larger than 8 nm. Suboptimal conditions, like i.e., too slow (3.0 mL h$^{-1}$) or too high (36 mL h$^{-1}$) cluster injection rates, or improper cluster precursor concentration, led to secondary nucleation and interparticle ripening (Supplementary Fig. 17). The effects of the different reaction conditions are summarized in Supplementary Table 3.

By optimizing these parameters, as detailed in Supplementary Table 4 and Supplementary Fig. 18, and by introducing injection-annealing cycles (see Fig. 1c), we directed the growth of non-agglomerated, large InAs QDs. Specifically, an injection rate of 6 mL h$^{-1}$ was used for adding of 4.5 mL of InAs cluster solution (1.1 mg mL$^{-1}$) into 6.0 mL of InAs seed solution (1.3 mg mL$^{-1}$), attributing to one injection step, followed by a 30-min annealing step at 300 °C. This process was repeated up to 16 times (8 injection and 8 annealing steps, or 8 growth cycles) and could be quenched after each step depending on the desired QD size. We refer to this as our standard seeded growth synthesis. The samples were purified post-synthetically using size-selective precipitation techniques (hereafter referred to as 'precipitation techniques', see Methods). As a result, each growth step contributed to the increase in QD size, as observed in the fraction containing the largest QDs, as summarized in Supplementary Table 5.

Figure 5 summarizes the data of the resulting InAs QDs after applying the precipitation techniques. From the absorption spectra (Fig. 5a), multiple absorption features corresponding to excitonic

intraband transitions[45] were observed. For samples with the first absorption peak below ≈2000 nm, well-resolved absorption features were noted, along with the expected redshift upon growth. However, the spectra of the larger QDs showed lower quality due to their poor solubility at RT The absorption edge for the largest QDs (produced after 8 growth cycles) shifted deep into the SWIR region, reaching up to ≈2300 nm (≈ 0.54 eV).

In Fig. 5b, we compared the experimental QD sizes and corresponding first absorption maxima with values calculated using Eq. (1) provided by Kuno et al.[46], which relates the size $d$ of InAs QDs to the energy $E_g$ of their first excitonic transition. This correlation is a slightly modified version of the one proposed by Yu et al.[47]. Both experimental and calculated values are provided in Supplementary Table 5 and are plotted in Fig. 5b. The good agreement between the experimental and calculated plots indicated that the $E_g$-$d$ relationship, originally derived for InAs QDs with sizes up to 6 nm[48], is also applicable to larger InAs QDs with sizes at least up to 18 nm.

$$E_g = 0.29\,\text{eV} + \frac{4.96\,\text{eV nm}^{-1}}{d} - \frac{2.24\,\text{eV nm}^{-2}}{d^2} \qquad (1)$$

TEM and HR-TEM images provided in Fig. 5c–f showed InAs QD samples with low size dispersity and up to 18 nm in size. Larger QDs exhibited a preferred octahedral and "fortuna cookie" morphologies, while smaller ones exhibited a preferred spherical and tetrahedral shapes. Before size selection, QDs obtained, e.g., in a 16-step synthesis had an average size of 14 nm (Supplementary Fig. 19). XRD patterns (Supplementary Fig. 20) confirmed the cubic zinc blende crystal structure of InAs, showing that the seeded growth synthesis provided highly crystalline InAs QDs.

To access the role of annealing steps in our seeded growth method, we conducted a control experiment where cluster solution was continuously injected without any annealing steps in between (Supplementary Table 6). Also here, non-agglomerated large InAs QDs were produced, but the absorption edge reached only 1750 nm after injection of the same amount (22.5 mL) of cluster precursor solution, compared to ≈1940 nm for the standard seeded growth method. The smaller QD size in the control experiment suggests that the annealing steps were crucial for promoting further QD growth, likely due to the lower reactivity of the cluster precursors. However, continuous injection without annealing could be advantageous for adapting the process to continuous flow reactors, with spatially separated nucleation and growth.

## Seeded growth synthesis utilizing diluted seeds

To target the growth of even larger InAs QDs, we diluted the seed solution by a factor of five and performed a synthesis procedure similar to our standard seeded growth method, involving injection-annealing cycles. According to classical nucleation and growth theory[28], dilution of the seed solution, where all seeds compete for the same amount of precursors, should promote the growth of larger QDs. Details of the synthesis parameters are provided in Supplementary Fig. 21, Supplementary Table 7, and in the Methods.

Already after the first injection of 4.5 mL of cluster solution, we achieved InAs QDs showing an absorption edge extending up to ≈1600 nm, compared to ≈1150 nm produced by the standard method with a fivefold higher seed concentration. Following annealing, the QDs grew so large that they became colloidally unstable in tetrachloroethylene (TCE) or toluene at RT, making them unsuitable for absorption measurements in solution. After 5 injection-annealing cycles (10 synthesis steps), InAs QDs with average sizes up to 36 nm (post-precipitation) were obtained; representative TEM images are provided in Fig. 6a, b. Before size-selection, the QDs in the sample obtained after 10 synthesis steps had an average size of 24 nm (see Supplementary Fig. 22). Compared to the standard seeded growth

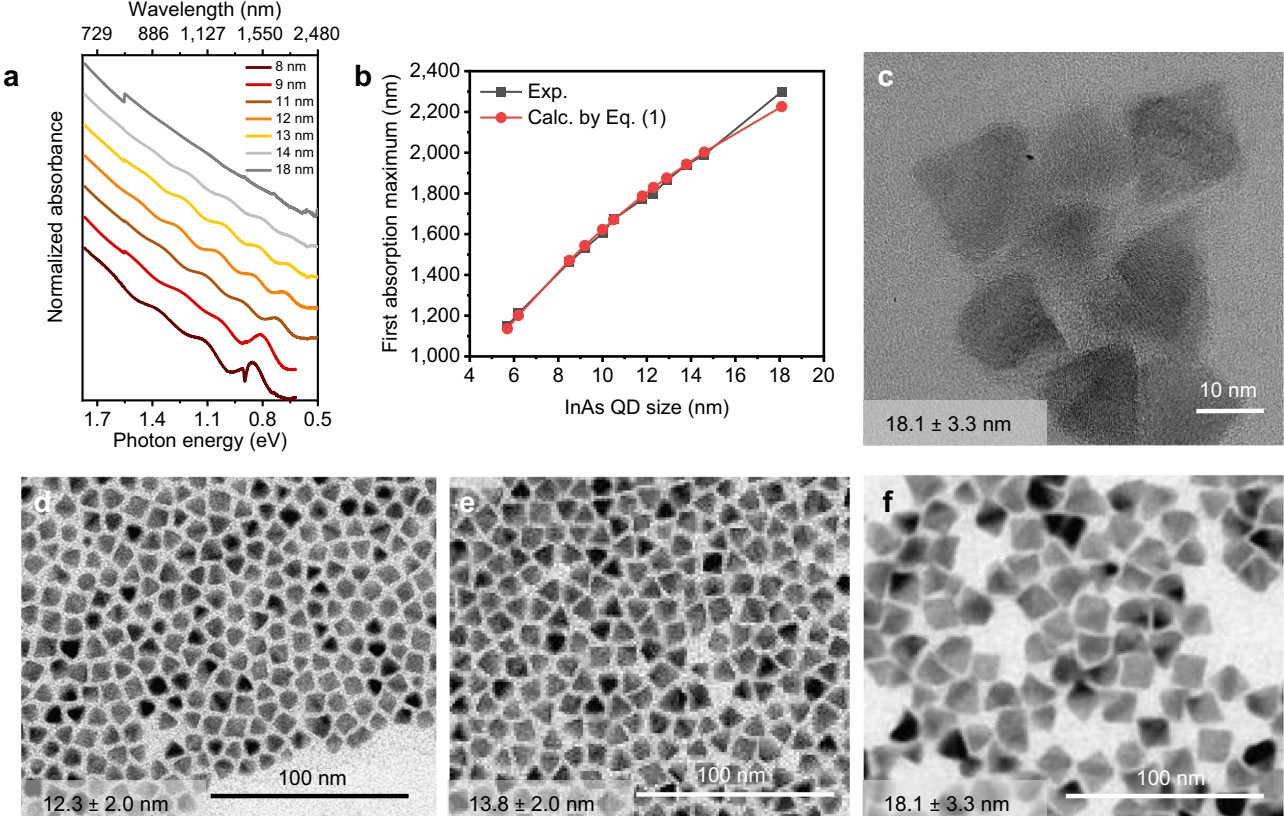

**Fig. 5 | InAs QDs synthesized via the standard seeded growth method using diluted seeds and clusters as precursors, after size-selective precipitation techniques were applied**[41,42]**. a** Size-dependent absorption spectra of representative InAs QD fractions, displayed with an offset and color-coded by size. Spectra were normalized to [0, 1]; four spectra (9 nm, 11 nm, 13 nm, 18 nm) were previously baseline-corrected. **b** Size-dependence of the first absorption maxima of InAs QDs, with experimental data (black) and calculated values (red) using the procedure from ref. 46, both listed in Supplementary Table 5. **c** Representative HR-TEM image, and **d**–**f** representative TEM images of large InAs QDs. TEM images in (**d**, **e**) are from experiments independently repeated three times, while those in **c**, **f** are from an experiment performed once. Source data are provided as a Source Data file.

approach (Supplementary Fig. 19), the size distribution became broader (46% vs. 34%), and a bimodal size distribution could be observed. Nevertheless, the experiment revealed alternative InAs QD morphologies, including icosahedrons and 8-edged polyhedrons. The representative XRD diffractogram of the largest InAs QDs (≈ 36 nm) in Fig. 6c confirmed the cubic zinc blende crystal structure and showed high crystallinity.

**Seedless growth synthesis of large InAs quantum dots**

To further investigate the effect of seeds on QD size, we reduced the seed concentration to zero, replacing the seed solution with an equivalent volume of 5.9 mL OAm and 0.1 mL TOP. We refer to this method as seedless growth (see Fig. 1c).

For the seedless growth, all other reaction parameters were kept identical to those of the standard seeded growth approach (see 'Methods'. Remarkably, by applying this technique, we were able to produce even larger InAs NPs (Fig. 7a–d). Supplementary Table 8 summarizes the NP sizes achieved after each synthesis step. After 5 injection-annealing cycles (10 synthesis steps), we achieved InAs NPs with average sizes up to 40 nm, with some individual particles reaching 55–65 nm (Supplementary Figs. 23 and 24). A representative HR-TEM image is shown in Supplementary Fig. 25. The as-received NPs (prior to size-selection) had an average size of 34 nm and a bimodal size distribution (see Supplementary Fig. 26). Notably, the size distribution decreased compared to the seeded growth method utilizing diluted seeds, both before size selection (35% vs. 46%). The QDs exhibited morphologies similar to those observed in seeded growth syntheses, including tetrahedral, octahedral, 8-edged polyhedral, and 'fortuna

cookie' shapes[49,50]. Notably, the average size of the InAs NPs produced in this seedless growth method (after precipitation techniques were applied) was more than three times larger than the largest colloidal, non-elongated InAs QDs reported in ref. 34.

SAED analysis (Fig. 7e) and XRD diffraction pattern (Supplementary Fig. 27) confirmed the crystalline nature of the reaction product, consistent with the cubic zinc blende crystal structure of bulk InAs. SEM (scanning electron microscope) images of 40 nm large NPs dropcasted and dried on silicon wafers showed favorable stacking behavior (Supplementary Fig. 28) and highlighted their potential for self-assembly into structures with long-range order in two or three dimensions. These characteristics are particularly promising for applications such as the development of absorber layers in optoelectronic devices.

Absorption spectra of large, colloidally unstable InAs NPs in TCE (Supplementary Fig. 29a) showed absorption in the range up to ≈2600 nm, but without distinct absorption maxima. Beyond ≈2600 nm, scattering from NPs and overlap with ligand absorption interfered with the measurements.

Absorption measurements using an integrating sphere were not feasible in the MIR range due to technical limitations. Therefore, additional measurements were conducted on a substrate (Supplementary Fig. 29b). However, scattering effects and ligand absorption features persisted above ≈2600 nm. Despite the absence of distinct absorption maxima, a redshift in the absorption edge was observed, suggesting quantum confinement effects. In general, QDs sized between the electron Bohr radius (28.5 nm for InAs)[51] and the exciton Bohr radius (30–44 nm for InAs)[10,51,52] are expected to fall into the weak

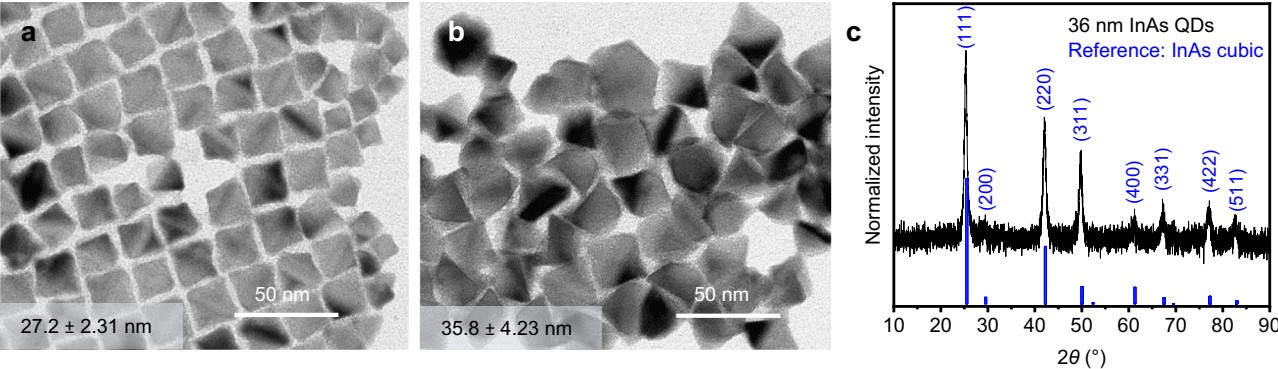

**Fig. 6 | InAs NPs synthesized via the seeded growth method using diluted seeds and clusters as precursors, with size-selective precipitation techniques applied**[41,42]. **a, b** TEM images of different InAs NP samples, synthesized once. **c** XRD diffractogram of the largest InAs NPs, normalized to [0, 1], with the InAs reference shown in blue. Source data are provided as a Source Data file.

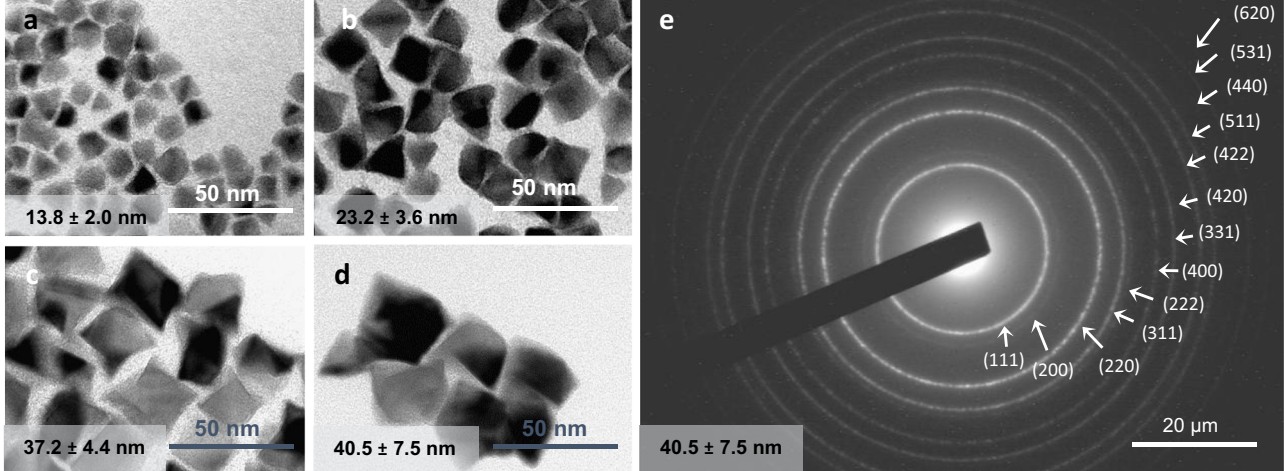

**Fig. 7 | InAs NPs synthesized via the seedless growth method using clusters as precursor, with size-selective precipitation techniques applied**[41,42]. **a–d** Representative TEM images of InAs NP samples; experiments were independently repeated twice. **e** SAED pattern of the largest InAs NPs.

confinement regime, where absorption maxima are typically absent[51,53]. This means, in our case, that probably, no distinct absorption maxima would be observed for large InAs NPs in this size regime.

An additional solid-phase absorption spectrum acquired from a thin film of deposited QDs using attenuated total reflection infrared (ATR-IR) spectroscopy is presented in Supplementary Fig. 29c. The broad absorption feature of the OAm ligands' –NH$_2$ group (3500–3350 cm$^{-1}$)[54,55] overlapped with the expected NP absorption edge ($\approx$ 3020 nm, extrapolated from the equation in ref. 46, or up to $\approx$3444 nm, according to InAs bulk bandgap, assuming these large NPs are QDs).

Further ATR-IR analysis of InAs NP samples with average sizes ranging from 12 to 36 nm (Supplementary Fig. 29d, e) revealed, in addition to the ligand features, a size-dependent, broad absorption band in the mid-infrared (MIR) range (1600–920 cm$^{-1}$), likely associated with the vibrational modes of OAm[56]. As the QD size increased, this band shifted further into the red. Such a size-dependent effect is particularly interesting and likely arises from the considerable variation in size within the same material, which amplifies this effect.

Supplementary Fig. 29e presents a zoom-in of the ATR-IR spectra in Supplementary Fig. 29d. Arrows mark the corresponding first absorption peaks, calculated using the procedure from ref. 46. The measurements revealed a qualitative redshift in absorption as the NP size increased. However, no distinct absorption features were observed, as expected, since ATR-IR characterizes the NP surface.

Given the ongoing debate surrounding the exciton Bohr radius of InAs QDs (ranging from 30 to 44 nm, as mentioned above)[10,51,52], it remains uncertain whether InAs NPs larger than 30 nm can be classified as QDs. Even if they do, they likely would not exhibit distinct absorption bands, as they would fall into the weak confinement regime. Instead, they would be better classified as near-bulk QDs[51]. Due to the addressed wavelength range, an extensive optical characterization of the large InAs particles' properties requires further improvements. This characterization is within the scope of follow-up studies, where measurements on the near-bulk QDs could provide insights into the exciton Bohr radius of InAs QDs, a fundamental parameter in the study of QD systems.

We also performed initial experiments to investigate the mechanism behind the seedless growth process. As detailed in Supplementary Note 2, we observed a delay of several minutes after the start of cluster precursor supply, indicated by the color change of the reaction solution (Supplementary Fig. 30). We attribute this delay to the dissolution and monomer release process of the clusters, during which the concentration of free In$^{3+}$ and As$^{3-}$ monomers increased as cluster precursors were introduced. Once the critical concentration was reached, nucleation of small InAs QDs (seeds) began. These QDs continued to grow, forming larger particles. Due to the relatively low number of these naturally formed seeds, the final particles were larger compared to those obtained using the same amount of cluster precursor in the seeded growth method. However, since these seeds could

form at different times during the reaction (continuous nucleation), the growth period for the final particles varied, leading to a broader distribution in particle size.

These findings are supported by absorption spectra taken during the cluster injection process (Supplementary Fig. 31, black and red lines). We observed an increase in concentration over time, accompanied by higher absorbance at longer wavelengths, indicating the growth of larger nanostructures. A sharp absorption maximum at ≈350 nm indicated the presence of ultra-small, naturally formed InAs seeds (Supplementary Fig. 31, black line). This peak slightly broadened with further cluster supply (Supplementary Fig. 31, red line). Comparing this to the spectra of early-stage structures from Fig. 3c (also shown in Supplementary Fig. 31, dashed green and violet lines), we observed a ≈320 nm minimum, where the early-stage structures exhibited a second-order maximum. This suggests that early-stage structures were not present in these aliquots. In addition, there was no overlap with the band offset (532 nm) of the initial cluster solution (Supplementary Fig. 31, blue double line), further emphasizing the dissolution and conversion process of the clusters.

To minimize size broadening, future studies should focus on developing strategies to better control the nucleation process. Possible approaches include reducing the initial volume of hot solvent, adjusting the cluster injection rate immediately after nucleation, and optimizing the cluster concentration.

In a control experiment, no annealing steps were conducted. Consistent with the control-seeded growth experiment, non-aggregated, large InAs QDs were still produced. After injecting the same volume of cluster solution (22.5 mL) as in the standard seedless growth approach, the resulting QDs were 27 nm in size, compared to 40 nm when annealing steps were included (after precipitation techniques were applied). This suggests that alternating annealing steps play a crucial role in the growth process when using clusters as precursors.

In conclusion, our work demonstrates colloidal synthesis strategies for InAs NPs across a broad size regime, from clusters to near-bulk QDs, overcoming long-standing challenges in the growth of large III-V nanocrystals. Our 'green' synthesis approach introduces a versatile, scalable platform to deliver long-term stable InAs clusters as precursors, enabling both the seeded growth and the seedless growth pathways. This flexibility allows tuning of size and optical properties, with access to NPs reaching (and possibly exceeding) the exciton Bohr radius, depending on its 'true' value, which is still under discussion.

Beyond enabling access to large InAs QDs reaching near-bulk values, our method offers a more sustainable alternative to traditional organometallic routes[10,26,32], combining the use of less hazardous and cheaper, commercially available chemicals[26] with involving clusters as precursors[13,39,40]. The issue with metallic In formation[26] was addressed by removing the byproducts of cluster and/or seed synthesis prior to their use in subsequent seeded or seedless growth. The strategy is compatible with continuous flow reactors, enabling large-scale production of RoHS-compliant Pb- and Hg-free colloidal QDs for the NIR and SWIR regions. By extending the accessible size range and optimizing the synthetic toolbox for InAs NPs, this work provides a basis for studies and applications in IR optoelectronics, including biophotonics, telecommunications, and sensing.

## Methods
### Materials
Indium(I) chloride (InCl, anhydrous, PURATREM, 99.99%), tetrachloroethylene (TCE, anhydrous, Th. Geyer, ≥99%), tri-n-octylphosphine (TOP, Strem Chemicals, 97%), tris(dimethylamino) arsine (TDMAAs, amino-As, Strem Chemicals, 99%), liquid paraffin oil (Th. Geyer), toluene (anhydrous, Sigma-Aldrich, 99.8%), and ethanol (anhydrous, VWR, ≥99.8%) were used as received. Oleylamine (OAm, Merck, 98%) was dried at 80 °C under vacuum for 1.5 h. All chemicals were stored and handled in an inert nitrogen

atmosphere within a glove box, where OAm was kept refrigerated. Caution: Although amino-As is considered as a 'greener' precursor, it is highly toxic and should only be handled inside a glove box under an inert atmosphere.

### Synthetic procedures
All synthesis, purification, and sample preparation procedures were conducted inside a nitrogen-filled glove box equipped with a Schlenk line and a Minispin centrifuge. Larger volumes were centrifuged outside the glove box in hermetically sealed tubes.

### Arsenic precursor solution for cluster synthesis
The arsenic precursor was prepared in situ following a procedure from ref. 26 with modifications, using 150 μL of amino-As in 2.0 mL of OAm. The solution was heated to 50 °C and stirred for at least 15 min, until no bubble formation was observed.

### Synthesis of InAs cluster precursors
For the synthesis of InAs cluster precursors, we used a procedure from ref. 26 with modifications. A mixture of InCl (0.300 g, 2.00 mmol, 3.1 eq.) and dry OAm (28.8 mL) was degassed at RT for 10 min. After switching to nitrogen, 1.2 mL of TOP was injected, and the mixture was heated to 110–180 °C. The arsenic precursor solution (1.6 mL, 0.64 mmol As, 1 eq.) was then added via hot injection. The reaction was maintained for 2 min to 3 h 20 min at a constant temperature, depending on the desired average cluster size (Supplementary Table 1). The synthesis was scaled down proportionally if smaller volumes of cluster precursor solution were sufficient. Insoluble byproducts were removed by sedimentation or centrifugation.

Clusters used as precursors in the seeded and/or seedless growth approaches were synthesized at 130 °C for 76 min. The reaction yield of these clusters was 49% based on As and 16% based on In, as determined by thermogravimetric analysis (TGA), with a concentration of 1.9 mg mL$^{-1}$.

The reaction yield was estimated as follows. First, the reaction mixture was centrifuged to separate insoluble byproducts. Then, 3.0 mL of 36.1 mL original red supernatant containing InAs clusters (stock solution) was washed with ethanol. The resulting pellet was used for the TGA. With 20.26% of inorganic compounds, this resulted in 5.7 mg of InAs clusters. Since the original solution was 31.6 mL, this corresponded to 59.99 mg (0.316 mmol) of InAs clusters. For further details, see Supplementary Note 3 and Supplementary Table 9.

### Synthesis of InAs seeds
InAs seeds were synthesized using a heat-up method. In our standard procedure, InAs clusters were first synthesized at 180 °C for 15 min, as described above. The reaction mixture was then heated to 300 °C and immediately cooled to RT using a paraffin oil bath. Insoluble byproducts were removed by sedimentation or centrifugation. The reaction yield of the resulting InAs seeds was 90% based on As and 23% based on In, as determined by TGA. The concentration of seeds was 3.3 mg mL$^{-1}$.

To estimate the reaction yield, the reaction mixture was first centrifuged to remove any insoluble byproducts. Then, 900 μL of the supernatant containing InAs seeds were washed with ethanol, and the resulting pellet was used for the TGA. The sample contained 54.65% inorganic residuals, which corresponded to 3.1 mg of inorganic InAs material. For a total volume of 23.7 mL of the stock seed solution, it resulted in 81.6 mg (0.430 mmol) of InAs. For further details, see Supplementary Note 4 and Supplementary Table 10.

### Seeded growth synthesis of large InAs QDs
Large InAs QDs were synthesized via a seeded growth method using InAs seeds and InAs cluster precursors. Prior to use, the cluster precursor and seed mixtures were separated from insoluble byproducts by centrifugation.

To ensure reproducible reaction conditions, the sizes and concentrations of seeds and clusters were monitored by absorption spectroscopy using the position and magnitude (optical density, OD) of the excitonic absorbance maximum (see Supplementary Tables 4 and 7). Concentrations were equalized by dilution with dry OAm to obtain the target OD values. For the standard seeded growth approach, the seed concentration was adjusted to 1.3 mg mL$^{-1}$ (OD 0.16 at the absorption maximum of 931 nm), whereas for the diluted seed approach, it was adjusted to 0.24 mg mL$^{-1}$ (OD 0.034). In both cases, a total seed solution volume of 6.0 mL was used. The InAs cluster concentration was set to 1.1 mg mL$^{-1}$ (OD 0.22 at the absorption maximum) for both approaches. Depending on the number of synthesis steps, 4.5–36.0 mL of InAs cluster solution was loaded into a syringe.

A 6.0 mL volume of InAs seed solution was transferred to a three-neck round-bottom flask and heated to 300 °C under nitrogen flow. Upon reaching 300 °C, 4.5 mL of InAs cluster solution (corresponding to 4.9 mg of InAs) was continuously injected into the seed solution using a syringe pump at a rate of 6 mL h$^{-1}$. This was followed by an annealing step of 30 min at the same temperature. The injection-annealing cycle was repeated multiple times to achieve the desired QD size (see Supplementary Table 5). A control experiment was performed without the annealing steps (Supplementary Table 6).

During quenching of the reaction in a paraffin oil bath, the black reaction mixture turned brown and became unstable, which was attributed to the presence of large InAs QDs that are colloidally unstable in OAm at lower temperatures.

InAs QD fractions with a low polydispersity were isolated using size-selective precipitation with anhydrous toluene as the solvent and ethanol as the non-solvent. The reaction mixture was first shaken and centrifuged to separate QDs with sizes above ≈9 nm, which were colloidally unstable in OAm at RT but stable in toluene and TCE for sizes up to ≈19 nm. If a pellet formed, it was dispersed into toluene or TCE, and centrifuged again to separate QDs larger than ≈19 nm. In the final step, if a pellet formed, it was dispersed in TCE or toluene, and ethanol was added dropwise until turbidity was observed. The mixture was centrifuged, and the resulting pellet was dispersed in TCE for characterization. If no pellet formed, the supernatant containing QDs was subjected to size-selective precipitation using ethanol as the non-solvent. A rainbow-like shimmer observed in regions of thin pellet thickness was attributed to interference effects, indicating a narrow NP size distribution.

For determining the reaction yield of the 16-step standard seeded growth synthesis, 500 μL of the reaction mixture was washed with ethanol, and the resulting pellet was used for the TGA. With 11.52% of inorganic material, this resulted in 3.7 mg of InAs. For 42 mL of the original reaction mixture, this corresponded to 36 mg (0.19 mmol) of InAs. For calculations of the reaction yield, see Supplementary Note 5.

## Seedless growth synthesis of large InAs QDs

The seedless growth method was conducted analogously to the standard seeded growth approach, with the only difference being that an OAm-TOP solution (5.9 mL OAm and 0.1 mL TOP) was used instead of the seed solution (6.0 mL), see Supplementary Table 8. In a control experiment, no annealing steps were included. Purification was performed as described for the seeded growth approach. After storage of the reaction product obtained from 10 synthesis steps at RT without agitation, the supernatant became completely clear and colorless, indicating that all synthesized NPs, likely larger than ≈9 nm, had sedimented.

For determining the reaction yield of the 10-step seedless growth synthesis, 1000 μL of the reaction mixture after shaking was washed with ethanol and dispersed in 100 μL of TCE, from which 25 μL was used for the TGA, resulting in 6.15% inorganic material (3.5 mg InAs). For 28.5 mL of the final reaction solution, this results in 24.5 mg (0.129 mmol) of InAs. For determining the reaction yield of the 6-step

seedless growth synthesis, 45 μL of the reaction mixture was used for TGA without further purification, with 0.09% inorganic material (0.0392 mg) in 43.6 mg of a liquid sample containing non-evaporable OAm and TOP. In a total reaction volume of 19.5 mL, it corresponds to 17.0 mg (0.0896 mmol) of InAs. For calculations, see Supplementary Note 6.

## Optical absorption spectroscopy

Absorption measurements were performed using a Cary 5000 UV-Vis-NIR spectrophotometer (Varian) in a dual-beam mode. Sample preparation was carried out inside a glove box.

For absorption measurements in solution, the sample was placed in a 1.00-mm-thick quartz cuvette from Hellma and hermetically sealed. After measurements, the sample was returned to the glove box. For measurements up to ≈2300 nm, 110-QS High Performance cuvettes (200–2500 nm) were used, whereas for measurements in the MIR range, 110-QX Extended Range cuvettes (200–3500 nm) were utilized.

Sample preparation for absorption measurements on substrates was conducted inside a glove box. Purified QDs were deposited as a thin film by drop-casting from TCE or toluene onto a 25 mm diameter water-free fused silica (SiO$_2$) substrate (Andover Corporation). After solvent evaporation, measurements were performed under ambient conditions.

## ATR-IR spectroscopy

ATR-IR measurements were conducted using a Golden Gate ATR System from Bruker Optik GmbH (6000 –600 cm$^{-1}$). Prior to measurement, the QD sample was repeatedly precipitated and redispersed in toluene inside a glove box. The resulting dispersion was drop-cast onto the ATR crystal under ambient conditions, allowing the solvent to evaporate. This procedure was repeated until a sufficiently thick film had formed on the crystal surface. Before sample acquisition, the background spectrum was recorded and automatically subtracted from the sample spectrum.

## TEM and SAED

TEM images and SAED patterns were acquired using a JEOL JEM 1011 transmission electron microscope equipped with a LaB$_6$ cathode operating at an acceleration voltage of 100 kV, and a SIS CCD camera system (1376 × 1032 pixels). The purified and diluted sample was drop-cast onto a carbon-coated copper grid (Science Services) under normal conditions, allowing the solvent to evaporate.

## HR-TEM and HAADF-STEM

HR-TEM and HAADF-STEM samples were prepared under a nitrogen atmosphere using the same procedure as for TEM sample preparation. Images were obtained using JEOL JEM−2200FS equipped with a field emission gun.

## SEM

For SEM analysis, the purified and concentrated QD sample was drop-cast onto a silicon wafer under ambient conditions, followed by solvent evaporation. SEM was performed using an FEI Quanta 3D FEG instrument equipped with a field emission gun.

## XRD

XRD sample preparation was carried out inside a glove box. A purified dispersion in toluene was drop-cast onto a silicon wafer, and the solvent was allowed to evaporate. Then, the resulting film was optionally covered with an insulating Kapton® polyimide tape to prevent exposure to oxygen and moisture. Measurements were performed using either a Panalytical MPD X'Pert Pro diffractometer or a Philips X'Pert PRO MPD diffractometer, both equipped with a Cu $K_\alpha$ X-ray source, under continuous nitrogen flow. Backgrounds from empty wafers, or those covered with Kapton® polyimide tape, were subtracted from sample measurements.

## Elemental analysis
The sample was prepared inside a glove box by washing three times with ethanol and finally dissolving in toluene. Outside the glove box, the sample was digested at 250 °C in a microwave using nitric acid ($HNO_3$) and water, and subsequently diluted. The analysis was performed in triplicate to ensure reproducibility, using ICP-OES spectrometer ARCOS from Spectro.

## TGA
TGA was employed to determine the weight percentage of the inorganic component in the sample, which was then used to calculate the reaction yield. A measured volume of the liquid sample was placed into a TGA sample cup inside a glove box, allowing the solvent to evaporate at 80 °C. The mass of the remaining residue was subsequently recorded. TGA measurements were performed using a NETZSCH TG 209 F1 220-10-039-K thermogravimetric analyzer.

## Reporting summary
Further information on research design is available in the Nature Portfolio Reporting Summary linked to this article.

## Data availability
The data that support the findings of this study are available from the corresponding authors upon request. Unprocessed raw data generated in this study, including elemental analysis, TEM, HR-TEM, XRD, SEM, SAED, absorption and ATR-IR spectroscopy, have been deposited in the University of Hamburg ZFDM Repository[57]. A preprint of this work was made available on ChemRxiv[49] and was originally published as part of the doctoral dissertation[50] of Ekaterina Salikhova at Universität Hamburg in 2025. Source data are provided with this paper.

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

## Acknowledgements

The project has received funding from the European Union's Horizon 2020 research and innovation programme under grant agreement No 101016734. We thank the *Forschungswerkstatt der Physikalischen Chemie der Universität Hamburg* for production of the glove box components, Stefan Werner for SAED measurements and TEM images, Andrea Köppen, Robert Schön, and Dr. Charlotte Ruhmlieb for HR-TEM and SEM images, Isabelle Nevoigt and Dr. Frank Hoffmann for XRD measurements, Dr. Kathrin Hoppe and Prof. Dr. Nadja Bigall for providing access to the ATR-IR spectrometer, Prof. Dr. Jongwook Kim from CNRS for helpful discussion regarding the detection of plasmonic NPs, Dmitrij Lutz and Dr. Dirk Eifler for the elemental analysis, and Dr. Sören Becker for feedback and discussions.

## Author contributions

E.S. developed, designed, and carried out the experiments, analyzed the data, wrote and revised the manuscript. A.M., H.S., and J.S.N. supervised the project, provided feedback, and proofread the manuscript. J.S.N. and H.S. acquired the funding.

## Funding

## Competing interests

E.S., J.S.N., and A.M., together with Dr. Sören Becker, are inventors on a German patent application filed by *Fraunhofer-Gesellschaft zur Förderung der angewandten Forschung e. V.* with the German Patent and Trade Mark Office (DPMA, Munich, Germany; Application No. 10 2024 137 678.3, pending), including content reported in this paper. The remaining authors declare no competing interests.
