## [Transparent Peer Review file · Nature Communications]

Colloidal synthesis of large near-bulk InAs quantum dots through seeded and seedless growth using cluster precursors

Corresponding Author: Dr Ekaterina Salikhova

Version 0:

Reviewer comments:

Reviewer #1

(Remarks to the Author)

The manuscript from Salikhova et al. describes a method to synthesize very large colloidal InAs QDs. Given the relevance of such QDs for SWIR applications and the increasing interest from the research community in this class of QDs, I believe the manuscript could be worth of publication in Nature Communication. Nonetheless, the manuscript cannot be accepted in its current state, given the lack of cluster and qualitative discussion and lack of experimental details (seriously hampering the possibility to reproduce such synthetic approach in other labs). An extensive revision of the manuscript is necessary before it can be accepted for publication.

Please, see my comments and questions here below:

- In general, the authors should revise the manuscript to avoid qualitative words like "huge", "increased monodispersity". In addition, the text is sometimes hard to follow (especially the final parts); the authors should improve its clarity.
- The authors wrote: "The as-taken aliquots were grey-brown due to the presence of metallic indium and insoluble organometallic intermediates. After sedimentation, the real color of the cluster solution could be observed." The authors should show XRD or another analysis to confirm the nature of the sediment (an elemental analysis?).
- How was the size of the clusters estimated in Figure 3a? It seems impossible from low resolution TEM. The authors should carry out HAADF-STEM to understand shape and size.
- What is the reaction yield of the clusters (e.g., those shown in Figure 3a)?
- The authors wrote: "For example, the clusters synthesized at a temperature of 135 °C for 1 h (Fig. 3a) and then heated up to 300 °C (Fig. 3b) led immediately to the formation of spherical particles, but with a tendency to form `necks` between the QDs". How is it possible that the absorption spectrum is so well defined if necking and/or aggregation and precipitation are present? One would not expect to observe the exciton peak in these conditions.
- The authors wrote: "The TEM images of those samples revealed the presence of larger, darker appearing nanostructures (see Supplementary Fig. 8), most likely metallic indium nanoparticles. This assumes that the cluster growth in the first reaction step was not stoichiometric, but that there was still unreacted In precursor present, before the QD growth started." When heating the clusters up to 185°C and then increasing the temperature to 300°C what are the byproducts of the synthesis and what is the synthesis yield? What is the stoichiometry of the InAs QDs made with this route?
- How can the authors calculate the amount of starting InAs QDs and clusters to be used in the seeded-growth approach, considering that in both cases they have to remove byproducts?
- The authors wrote: "For the synthesis process, 4.5 mL of InAs cluster solution was slowly injected into a solution of InAs seeds, followed by 30-minute annealing steps. This process was repeated multiple times, while the reaction temperature

was maintained at 300 °C.” Please, explain this procedure in more details as it is written now it is impossible to understand it.

- The authors wrote: “By tuning the concentration and size of clusters and seeds, as well as the injection rate, we were able to control the growth process, i. e., we managed to avoid side reactions such as interparticle ripening or secondary nucleation.” Please, also here include the parameters used and add relevant experimental details (which concentrations and sizes were used? What are the results for non-optimized parameters, all the discussions are extremely lackluster and superficial).

- What does it mean, “a 16-step seeded growth synthesis”? Are the authors injecting 16 times the precursors? In addition, the clusters the authors inject, at what temperature are they prepared (do the authors centrifuge each of these 16 additions to avoid injecting metallic or precipitating stuff)?

- “Fig. 4a shows that we obtained InAs QDs with well-defined multiple absorption features” Please explain the absorption features, are these real electronic transitions?

- The authors wrote: “The absorption spectra of the samples with larger sizes are of less quality, because these particles were hardly soluble at room temperature”, then in the next line: “These results show that the huge non-agglomerated InAs QDs are highly monodisperse and highly crystalline.” Which one is which? This makes no sense.

- The authors wrote: “We compared our experimental results for QD sizes and first absorption maxima (Supplementary Table 3) with the values calculated using the formula provided by Kuno et al.[17], which is a slightly modified version of the formula from [18]. This formula is based on the relationship between the first exciton transition peak energy and QD size. The resulting plot (Supplementary Fig. 11) demonstrates that the formula, originally developed for InAs QDs with average sizes up to 6 nm[19], remains applicable to larger QDs with sizes at least up to ~ 18 nm.” Please, add a graph with all data points, like this is very hard to follow.

- The authors wrote: “In a control experiment, we also tested if the continuous injection of cluster solution without performing the annealing steps in between also leads to non-agglomerated huge InAs QDs. In fact, this procedure also worked, but the final absorption edge was only 1,750 nm after injecting 22.5 mL of cluster solution (see Supplementary Table 4). This shows that the annealing steps are highly beneficial for the QD growth from cluster precursors, most likely due to the low reactivity of the clusters.” Please explain what are these annealing steps.

Reviewer #2

(Remarks to the Author)

This paper describes a novel method for growing large InAs QDs up to around 40 nm with high crystallinity and good size control starting from non-pyrophoric precursors. It addresses the long-standing challenge of extending the absorption range of InAs QDs in SWIR from 1600 nm to much longer wavelengths, beyond 2600 nm. The main approach relies on the use of well-defined InAs clusters obtained at lower temperature (180°C), which are either grown to 4 nm nanocrystals used in a seeded-growth reaction, or directly applied in a heat-up approach. The samples were characterized using standard techniques, in particular absorption spectroscopy, X-ray diffraction and TEM, and the main reaction parameters were explored to optimize the synthesis methods. In the introduction and conclusion, a bigger accent could be put on the impact for SWIR applications, since it presents a core achievement of the work and there are not much other Pb- and Hg-free QDs that can reach this region.

All in all, this manuscript appears perfectly suitable for publication in Nature Communications, due to the originality of the approach and quality of the results, which open up new horizons.

Nonetheless there is a list of points which should be addressed:

1) The difference between seeds and clusters is a bit hard to follow, especially in the second part of the manuscript, I would suggest to elaborate in more details figure 1 with some key features of both. Fig. 1 could also be improved to make it visually more attractive.

2) What are the reaction yields for the cluster synthesis and for the seeded-growth synthesis? insoluble side-products such as metallic indium (p. 3) were observed. In other words, how much, approximately, of the initially used InCl transforms into clusters and how much into In(0)? Does the yield change with bigger particles? Or between seedless and seeded growth?

3) Did the authors also try the seedless growth without annealing steps?

4) In Fig.2 the T0 spectrum could be added.

5) The absorption spectra of the >20 nm nanocrystals could have been taken on solid samples by applying diffuse reflectance spectroscopy instead of liquid phase measurements in TCE. Given the unique size regime of the obtained samples, this information would have been quite interesting and would potentially allow to draw figure S11 (which I suggest to put into the main manuscript) up to 40 nm. Also, it would allow to give a more precise evaluation of the absorption range

covered by the obtained samples, the value of 2600 nm seems to be a rough estimation.

6) In SI, Fig 1, not sure if the evolution is related to the reaction proceeding over longer times (two years!), or particles being slowly oxidized by traces of oxygen that might be present in the inert atmosphere.

7) Experimental findings should not be discussed in the Conclusions but in the Discussion section. In particular, Fig. S16 (the RT evolution of clusters) is intriguing: the authors ascribe the sharp absorption features to the formation of metallic nanoparticles - are there any data supporting this hypothesis?

8) The last phrase of the conclusions says that the continuous injection approach could be easily adapted to continuous flow synthesis. How could this be practically achieved? It seems that the continuous injection of precursors to a given reaction volume is indeed one step difficult to implement in continuous flow (?)

9) Given the size distribution, the specification of 40.5 nm looks like excessive precision, I would suggest throughout the manuscript to mention that the method gives access to QDs of up to 40 nm size.

10) Fig. 2: the inscription on the vials should be removed and eventually be replaced by a real legend.

11) A 1:1 In:As ratio has been applied for the cluster synthesis. In principle a 3:1 ratio would be required to reduce As from +3 to -3, it should be discussed why 1:1 is used.

12) The use of the term "highly monodisperse" may not be appropriate for the clusters, as their absorption spectra are not that well defined showing a rather broad excitonic peak. For the TEM images a couple of size histograms would be helpful.

13) The particles in Fig. 3a with an absorption maximum at 556 nm are estimated to have a size of 2.3 nm. Is this consistent with the sizing curve of InAs QDs? They could be likely smaller (cf Fig. S2)

14) In the Introduction there is a lack of references and precision: use of vague phrases in the introduction ("extraordinary optoelectronic properties", "crucial role", "variety of materials", "harsher reaction conditions", "several disadvantages"). For example, while as stated no further developments of the In(I)Cl / aminonictogen method have been reported for InAs QDs, this synthetic route has been used for InP and InSb, which could be mentioned in the introduction.

15) On p. 5 it is stated that keeping the insoluble side-products from the cluster synthesis improves the quality of the final QDs leading to larger sizes and narrower size distributions. Do the authors have any hypothesis on the mechanisms behind this behavior?

16) Fig. S6: the reference diffraction pattern of InAs should be added. Idem Fig. S10

17) Fig. S7: why absorbance is given in arbitrary units?

18) In the Manuscript and Supplementary Information "heating up" should be termed "heat-up" throughout the text.

Reviewer #3

(Remarks to the Author)

Reviewer #4

(Remarks to the Author)

Please, see my attached report.

Version 1:

Reviewer comments:

Reviewer #1

(Remarks to the Author)

I thanks the authors for addressing all the reviewer's comments/concerns regarding this manuscript, it has greatly improved.

I believe the manuscript will be of great interest to readership of Nature Communication and the community working on colloidal InAs Quantum dots.

In my view, the revised manuscript can be accepted for publication.

Reviewer #2

(Remarks to the Author)

The authors have carefully addressed all points I raised, and I recommend the publication of this article in its present form. Once again I would like to point out the high importance of the presented results for this field.

Maybe one small final comment concerning the reaction yield calculations:

the molar quantity given (0.316 mmol, cf. Suppl. note 3) is not that of InAs clusters but that of InAs units and it does not appear to indicate the reaction yield according to In: due to the reaction mechanism, In has to be used with a 3:1 excess compared to As. Therefore, just giving the reaction yield according to As will be good enough.

Reviewer #3

(Remarks to the Author)

Reviewer #4

(Remarks to the Author)

The authors have carefully addressed the comments from all reviewers. The clarity of the text has been improved.

Response to the Reviewer #1

We sincerely thank the Reviewer for their insightful suggestions and valuable feedback. We have carefully addressed the points raised in the review and made the following revisions.

The manuscript from Salikhova et al. describes a method to synthesize very large colloidal InAs QDs. Given the relevance of such QDs for SWIR applications and the increasing interest from the research community in this class of QDs, I believe the manuscript could be worth of publication in Nature Communication. Nonetheless, the manuscript cannot be accepted in its current state, given the lackluster and qualitative discussion and lack of experimental details (seriously hampering the possibility to reproduce such synthetic approach in other labs). An extensive revision of the manuscript is necessary before it can be accepted for publication.

Please, see my comments and questions here below:

- In general, the authors should revise the manuscript to avoid qualitative words like “huge”, “increased monodispersity”. In addition, the text is sometimes hard to follow (especially the final parts); the authors should improve its clarity.

We sincerely thank the reviewer for this constructive comment.

In response to the reviewer’s suggestion, we have carefully revised the manuscript to replace subjective expressions and qualitative terms such as ‘huge’, ‘ultra-large’, and ‘increased monodispersity’ with more precise, quantifiable descriptions. This revision enhances the clarity and accuracy of our work. We have also revised the manuscript title for greater specificity.

Additionally, we have focused on improving the clarity and flow of the manuscript, especially in the final sections. We believe these revisions will address the concerns raised and hope that the updated manuscript now meets the expected standards.

We appreciate the reviewer’s valuable feedback and hope that the revised manuscript now meets the expected standards.

- The authors wrote: “The as-taken aliquots were grey-brown due to the presence of metallic indium and insoluble organometallic intermediates. After sedimentation, the real color of the cluster solution could be observed.” The authors should show XRD or another analysis to confirm the nature of the sediment (an elemental analysis?).

We thank the reviewer for this insightful suggestion.

To provide stronger evidence for the nature of the sediment, we have performed XRD analysis. The new data, now included in Supplementary Fig. 2, clearly indicate that the sediment consists predominantly of metallic indium, with minor contributions from In_2O_3 and InAs. The presence of traces of In_2O_3 is likely due to side reactions between the indium precursor and trace amounts of oxygen-containing impurities in oleylamine. No pure InCl was detected.

To clarify this in the manuscript, we have added the following discussion on p. 6 of the revised version: 'To investigate the nature of the insoluble byproducts formed during cluster growth, we performed X-ray diffraction (XRD) analysis (see Supplementary Fig. 2). The precipitate primarily contained metallic In, likely arising from the disproportionation of InCl, as reported previously^[26]. Traces of InAs and indium oxide (In_2O_3) were also detected. The formation of In_2O_3 could be attributed to side reactions with oxygen-containing impurities in OAm. No pure InCl was detected.'^[42]

- How was the size of the clusters estimated in Figure 3a? It seems impossible from low resolution TEM. The authors should carry out HAADF-STEM to understand shape and size.

We thank the reviewer for this valuable comment.

Due to small size of clusters, and the low contrast, we could only estimate the size in the first place, and we had estimated the size based on the longer axis of these elongated structures. We have now proceeded additional HR-TEM and also HAADF-STEM measurements (as suggested) to understand the shape and the size more carefully, and added the images to Supplementary Fig. 9. Also, from those images it can be seen that the clusters seem to form elongated aggregates – possibly during deposition on the TEM grids. Hence, we estimated the size based on the average width of these aggregates to be 1.4 nm, which has been added to Fig. 4a of the revised manuscript (corresponding to Fig. 3a in the original version).

- What is the reaction yield of the clusters (e.g., those shown in Figure 3a)?

We thank the reviewer for raising this point. We have now determined the reaction yield for the clusters prepared at 130 °C for 76 min, which we use as cluster precursors in both the seeded and seedless growth approaches. The **reaction yield of the clusters** was ~49% (related to As). This information has been added to the revised manuscript in the Methods section and Supplementary Note 3.

- The authors wrote: “For example, the clusters synthesized at a temperature of 135 °C for 1 h (Fig. 3a) and then heated up to 300 °C (Fig. 3b) led immediately to the formation of spherical particles, but with a tendency to form ‘necks’ between the QDs”. How is it possible that the absorption spectrum is so well defined if necking and/or aggregation and precipitation are present? One would not expect to observe the exciton peak in these conditions.

We thank the reviewer for this important point.

This is likely due to a small fraction of particles undergoing agglomeration. Another possible explanation is that necking occurred during the deposition of samples for TEM analysis, or as a result of the TEM imaging conditions. We have added these points to the revised manuscript (p. 9): ‘The absorption spectra of the particle solution exhibited a pronounced excitonic absorption peak (Fig. 4d, red line). However, a tendency to form ‘necks’ between the QDs was observed in the TEM images. This is likely due to a small fraction of particles undergoing agglomeration, the deposition process during TEM sample preparation, or specific imaging conditions.’

At this point we would like to address a discrepancy in the reported reaction temperature. The value of 135 °C was mistakenly recorded. After review, we confirmed that the actual reaction temperature was maintained at 130 °C throughout the experiment. This correction has been made in the revised manuscript (p. 9, Fig. 4).

- The authors wrote: “The TEM images of those samples revealed the presence of larger, darker appearing nanostructures (see Supplementary Fig. 8), most likely metallic indium nanoparticles. This assumes that the cluster growth in the first reaction step was not stoichiometric, but that there was still unreacted In precursor present, before the QD growth started.” When heating the clusters up to 185°C and then increasing the temperature to 300°C what are the byproducts of the synthesis and what is the synthesis yield? What is the stoichiometry of the InAs QDs made with this route?

We thank the reviewer for the questions.

Regarding the synthesis at 180 °C (not 185 °C, which appears to be a minor oversight in the reviewer’s comment), followed by heating to 300 °C, the synthesis yield of InAs QDs was ~90% (related to As). This value has been added to the revised manuscript (see Methods section and Supplementary Note 4), as requested.

On p. 11, we added a discussion regarding the nature of byproducts: ‘We analyzed the insoluble byproducts from the standard heat-up approach using TEM and XRD. From the TEM images (Supplementary Fig. 13), a mixture of polydisperse particles with undefined shapes, along with smaller spherical particles and darker particles, was observed. As in the case of the cluster synthesis, InAs, along with a small amount of In₂O₃ and metallic In, were identified from XRD (Supplementary Fig. 14).^{[42]’}

Insoluble byproducts of this reaction were determined by XRD and TEM analysis and the corresponding data have been added to the revised manuscript (Supplementary Fig. 13 and Supplementary Fig. 14). The byproducts consist of InAs aggregates with undefined shape, minor amounts of In₂O₃ (likely formed *via* side reactions involving oxygen-containing

oleylamine impurities), and small amounts of metallic In, consistent with incomplete precursor conversion and possible temperature-dependent equilibrium shifts. No pure InCl was identified.

The stoichiometry of the InAs seeds synthesized *via* the heat-up method was determined as 1.78:1, and the information added to the revised manuscript (Supplementary Table 2 and p. 11): ‘Elemental analysis of the isolated QDs gave an In:As ratio of 1.78:1 (Supplementary Table 2). The In-rich QDs were in agreement with previous report on InAs QDs synthesized from amino-As.^{[13]’}

- How can the authors calculate the amount of starting InAs QDs and clusters to be used in the seeded-growth approach, considering that in both cases they have to remove byproducts?

We thank the reviewer for raising this important point.

Despite the batch nature of our cluster and seed syntheses, the resulting reaction solutions after removing insoluble byproducts showed high reproducibility in absorption spectra (see Fig. 2a and Supplementary Fig. 15). In our study, we standardized the amount and size of clusters and seeds used for the seeded and seedless growth approaches based on the optical density (OD) and magnitude of the absorption maximum, according to the values reported in Supplementary Table 4 and Supplementary Tab. 7. Such OD-based normalization offers a practical alternative to mass-based concentration determination (see e.g., Kim et al., (2021) 12:3013 *Nat. Commun.*).

We also have determined the mass-based quantities of starting InAs clusters and seeds and their concentrations. These values are provided in response to the reviewer’s later question regarding cluster and seed concentration and size, and are also included in the revised Methods section, Supplementary Note 3, and Supplementary Note 4 for completeness.

We now added this more in-detail description to the Methods section: ‘Previously, the cluster and the seed mixtures were separated from insoluble byproducts *via* centrifugation.^[41,42] To ensure reproducible reaction conditions, the sizes and concentrations of seeds and clusters were monitored *via* absorption spectroscopy (position and magnitude (OD) of the excitonic absorbance maximum; see Supplementary Table 4 and Supplementary Table 7). To equal the concentrations, dry OAm was used as a buffer to dilute the clusters and/or seeds to obtain the target OD. The seed concentration was adjusted to 1.3 mg mL⁻¹ (OD 0.16 at the absorption maximum of 931 nm) for the standard seeded growth approach and 0.24 mg mL⁻¹ (OD 0.034 at the absorption maximum) for the diluted seed approach, with a volume of 6.0 mL in both cases. An InAs cluster concentration of 1.1 mg mL⁻¹ (OD 0.22 at the absorption maximum) was used for both approaches.’

OD-based reaction parameters are listed in Supplementary Tables 4 and Supplementary Table 7 for both the standard and the diluted seeds seeded growth approaches. Below we summarize the mass-based reaction parameters presented in Supplementary Notes 3 and 4, and in the revised Methods section:

Stock seed solution concentration (as-synthesized *via* heat-up and determined after removing insoluble byproducts; Supplementary Note 4): **3.3 mg/mL**, corresponding to an OD of 0.46 at absorption maximum.

Adjusted seed solution concentration for the **standard seeded growth** approach (Supplementary Note 4): **1.3 mg/mL**, corresponding to an OD of 0.16 at absorption maximum (Supplementary Table 4).

Adjusted seed solution concentration for seeded growth approach with **diluted seeds** (Supplementary Note 4): **0.24 mg/mL**, corresponding to an OD of 0.034 at absorption maximum (Supplementary Table 7).

Average sizes of InAs seeds used for the seeded growth approach are presented in Fig. 4f (4.0 nm).

Stock cluster solution concentration (as-synthesized at 130 °C for 76 min, after removing insoluble byproducts; Supplementary Note 3): **1.9 mg/mL**, corresponding to an OD of 0.39.

Adjusted cluster concentration used for seeded growth approaches (Supplementary Note 3): **1.1 mg/mL**, corresponding to an OD of 0.22 (Supplementary Table 4).

Average size of InAs clusters used for the seeded growth approach are presented in Fig. 2 and p. 6, with short and long axes measuring approximately 1 nm and 4 nm, respectively.

These additions clarify the experimental conditions under which our QDs could be reliably obtained, and we hope they adequately address the reviewer's concerns.

- The authors wrote: "For the synthesis process, 4.5 mL of InAs cluster solution was slowly injected into a solution of InAs seeds, followed by 30-minute annealing steps. This process was repeated multiple times, while the reaction temperature was maintained at 300 °C." Please, explain this procedure in more details as it is written now it is impossible to understand it.

We appreciate the reviewer's suggestion for further clarification. To address this, we have revised the manuscript to provide a more detailed explanation of the seeded growth process. We also introduced the term 'injection-annealing cycle' (p. 12) to define a complete process step.

In our revised version, we have clarified the procedure as follows: 'In the last stage of growing large InAs QDs *via* the seeded growth approach, we used InAs cluster and seed solutions after removing insoluble byproducts. Cluster solution was continuously injected into a solution of InAs seeds at a constant reaction temperature of 300 °C using a syringe pump.^[41,42] [...] Specifically, an injection rate of 6 mL h⁻¹ was used for adding of 4.5 mL of InAs cluster solution (1.1 mg mL⁻¹) into 6.0 mL of InAs seed solution (1.3 mg mL⁻¹), attributing to one injection step, followed by a 30-minute annealing step at 300 °C. This process was repeated up to 16 times (8 injection and 8 annealing steps, or 8 growth cycles) and could be quenched after each step depending on the desired QD size. We refer to this as our standard seeded growth synthesis.'

In the revised Methods section, we wrote: ‘The seed concentration was adjusted to 1.3 mg mL^{-1} (OD 0.16 at the absorption maximum of 931 nm) for the standard seeded growth approach and 0.24 mg mL^{-1} (OD 0.034 at the absorption maximum) for the diluted seed approach, with a volume of 6.0 mL in both cases. An InAs cluster concentration of 1.1 mg mL^{-1} (OD 0.22 at the absorption maximum) was used for both approaches. Depending on the number of synthesis steps, 4.5 to 36.0 mL of InAs cluster solution was loaded into a syringe. A 6.0 mL volume of InAs seed solution was then transferred to a three-neck round-bottom flask and heated to $300 \text{ }^\circ\text{C}$ under nitrogen flow.^[41,42]

As soon as $300 \text{ }^\circ\text{C}$ were reached, 4.5 mL of InAs cluster solution (4.9 mg InAs) was continuously injected into the seed solution using a syringe pump at an injection rate of 6 mL h^{-1} . A 30-min annealing step at the same temperature followed. The injection-annealing cycles could be repeated multiple times, depending on desired QD size (see Supplementary Table 5). A control experiment was performed without the annealing steps (Supplementary Table 6).’

- The authors wrote: “By tuning the concentration and size of clusters and seeds, as well as the injection rate, we were able to control the growth process, i. e., we managed to avoid side reactions such as interparticle ripening or secondary nucleation.” Please, also here include the parameters used and add relevant experimental details (which concentrations and sizes were used? What are the results for non-optimized parameters, all the discussions are extremely lackluster and superficial).

We thank the reviewer for requesting a more quantitative description. To address this, we have expanded the description to include the experimental conditions and results for non-optimized parameters.

We have added the description of non-optimized parameters to the main text (p. 12) and Supplementary Table 3. Non-optimized reaction conditions led to secondary nucleation and interparticle ripening. In order to show the resulting QDs, we added TEM images (Supplementary Fig. 17). We revised the text as follows: ‘Key parameters, such as cluster and seed size, concentration, and injection rate of clusters, were critical for obtaining non-agglomerated InAs QDs larger than 8 nm. Suboptimal conditions, like i.e., too slow (3.0 mL h^{-1}) or too high (36 mL h^{-1}) cluster injection rates, or improper cluster concentration, led to secondary nucleation and interparticle ripening (Supplementary Fig. 17). The effects of the different reaction conditions are summarized in Supplementary Table 3.^[42]’

- What does it mean, “a 16-step seeded growth synthesis”? Are the authors injecting 16 times the precursors? In addition, the clusters the authors inject, at what temperature are they prepared (do the authors centrifuge each of these 16 additions to avoid injecting metallic or precipitating stuff)?

We thank the reviewer for these insightful questions.

Regarding the term ‘16-step seeded growth synthesis’, it does not refer to injecting the precursors 16 times. Rather, it describes a sequence of 16 alternating synthesis steps (or 8 growth cycles), consisting of 8 cluster injections and 8 annealing steps. Specifically, each cycle involves injecting 4.5 mL of the InAs cluster solution (1.1 mg/mL), followed by a 30-minute

annealing step at 300°C. This process is repeated up to 16 times, with the injection and annealing steps alternating. No purification steps are performed between the steps.

To clarify, we have updated the manuscript (p. 12): ‘Specifically, an injection rate of 6 mL h⁻¹ was used for adding of 4.5 mL of InAs cluster solution (1.1 mg mL⁻¹) into 6.0 mL of InAs seed solution (1.3 mg mL⁻¹), attributing to one injection step, followed by a 30-minute annealing step at 300 °C. This process was repeated up to 16 times (8 injection and 8 annealing steps, or 8 growth cycles) and could be quenched after each step depending on the desired QD size.’

As for the InAs clusters used in the injections, they are prepared by heating at 130°C for 76 min. We have now updated the revised manuscript (p. 5-6): ‘For cluster synthesis, amino-As solution in oleylamine (OAm) was hot-injected into a mixture of InCl in OAm and trioctylphosphine (TOP) at 130 °C, and the reaction was maintained for 76 min.’ In the revised Methods section, we also provide this information: ‘Clusters used as precursors in the seeded and/or seedless growth approaches were synthesized at 130 °C for 76 min.’

Following this, the cluster reaction mixture is centrifuged to remove any insoluble byproducts, such as metallic In, which could interfere with the seeded growth synthesis. We have also added this clarification to the revised manuscript (p. 12): ‘In the last stage of growing large InAs QDs *via* the seeded growth approach, we used InAs cluster and seed solutions after removing insoluble byproducts.’

- “Fig. 4a shows that we obtained InAs QDs with well-defined multiple absorption features” Please explain the absorption features, are these real electronic transitions?

We thank the reviewer for this insightful question.

The multiple absorption features observed in Fig. 4a (now Fig. 5a in the revised manuscript) are indeed real electronic transitions. These features correspond to quantized excitonic interband transitions in the InAs QDs, arising from the size-dependent confinement of charge carriers in the nanocrystals.

Specifically, we refer to the work of Fang et al. [47], who reported up to five clearly resolved optical absorbance steps in ultrathin InAs quantum membranes (3–19 nm thick). These features were attributed to discrete interband transitions between subbands in the valence and conduction bands. Despite the difference in geometry (2D membranes vs. QDs), the physical origin of these features — quantum confinement — is the same. The observed similarity in spectral structure further supports the interpretation of our absorption features as real electronic transitions.

For our InAs QDs, we have observed up to five absorption maxima in QDs up to 18 nm in size, which aligns well with previous findings in the literature.

To provide additional clarity, we have included a brief explanation and an additional reference in the revised manuscript (p. 12): ‘From the absorption spectra (Fig. 5a), multiple absorption features corresponding to excitonic intraband transitions^[47] were observed.’

- The authors wrote: “The absorption spectra of the samples with larger sizes are of less quality, because these particles were hardly soluble at room temperature”, then in the next line: “These results show that the huge non-agglomerated InAs QDs are highly monodisperse and highly crystalline.” Which one is which? This makes no sense.

We thank the reviewer for pointing out this phrasing.

We have revised the text in the manuscript to more clearly distinguish between the quality of the optical spectra (first cited sentence) and the properties of the QDs determined from TEM and XRD analysis (second cited sentence).

The absorption spectra of the larger QDs showed light scattering due to their poor solubility, which affects the quality of the absorption spectra. We revised the text as follows (p. 8): ‘For samples with the first absorption peak below ~2,000 nm, well-resolved absorption features were noted, along with the expected redshift upon growth. However, the spectra of the larger QDs showed lower quality due to their poor solubility at r. t.’

Despite the lower quality of absorption spectra due to poor solubility, the larger QDs were non-agglomerated, monodisperse (after size-selection), and crystalline (as originally stated), as observed from the TEM images (as originally stated). We avoided using the qualitative term ‘highly monodisperse’ to maintain objectivity in our description, and underlined the application of size-selective precipitation to obtain monodisperse fractions. We revised the text for clarity as follows (p. 13): ‘The samples were purified post-synthetically using size-selective precipitation techniques (hereafter referred to as ‘precipitation techniques’), as described in the Methods section. [...] XRD patterns (Supplementary Fig. 20) confirmed the cubic zinc blende crystal structure of InAs, showing that the seeded growth synthesis provided highly crystalline InAs QDs.^{[41,42]’}

And later on (p. 13): ‘TEM and HR-TEM images provided in Fig. 5c-f showed monodisperse InAs QD samples up to 18 nm in size.’

We hope these changes address the reviewer’s concerns.

- The authors wrote: “We compared our experimental results for QD sizes and first absorption maxima (Supplementary Table 3) with the values calculated using the formula provided by Kuno et al.[17], which is a slightly modified version of the formula from [18]. This formula is based on the relationship between the first exciton transition peak energy and QD size. The resulting plot (Supplementary Fig. 11) demonstrates that the formula, originally developed for InAs QDs with average sizes up to 6 nm[19], remains applicable to larger QDs with sizes at least up to ~ 18 nm.” Please, add a graph with all data points, like this is very hard to follow.

We thank the reviewer for the valuable suggestion.

We would like to point out that the requested graph with all data points was already included as Supplementary Fig. 11 (Fig. 5b in the revised version). However, we acknowledge that the original wording in the manuscript may not have clearly communicated that this plot contains both the calculated values (in red) based on the empirical formula from Kuno et al. [48] and

the experimentally measured data points (in black) derived from Supplementary Table 3 (Supplementary Table 5 in the revised manuscript).

To address this, we have clarified the manuscript text and moved the plot to the main text for better visibility (p. 14). The updated text (p. 13) reads: 'In Fig. 5b, we compared the experimental QD sizes and corresponding first absorption maxima with values calculated using the formula provided by Kuno et al,^[48] which relates the size of InAs QDs to the energy of their first excitonic transition. This equation is a slightly modified version of one proposed by Yu et al.^[49] Both experimental and calculated values are provided in Supplementary Table 5 and are plotted in Fig. 5b. The good agreement between the experimental and calculated plots indicated that the formula, originally developed for InAs QDs with sizes up to 6 nm^[50], is also applicable to larger InAs QDs with sizes at least up to 18 nm.^[41,42]'

- The authors wrote: "“In a control experiment, we also tested if the continuous injection of cluster solution without performing the annealing steps in between also leads to non-agglomerated huge InAs QDs. In fact, this procedure also worked, but the final absorption edge was only 1,750 nm after injecting 22.5 mL of cluster solution (see Supplementary Table 4). This shows that the annealing steps are highly beneficial for the QD growth from cluster precursors, most likely due to the low reactivity of the clusters.” Please explain what are these annealing steps.

We thank the reviewer for this important question.

The 'annealing step' refers to a 30-minute holding period at 300°C that follows the injection of 4.5 mL of InAs cluster solution. During this time, the newly added clusters are allowed to fully integrate into the growing InAs QDs, promoting further growth.

In the control experiment, continuous cluster injection was applied without the annealing steps. As a result, the QDs grew less effectively, as indicated by the lower final absorption edge (1,750 nm) compared the standard procedure (~1,940 nm). This demonstrates the importance of the annealing steps for optimal QD growth, likely due to the low reactivity of the clusters.

We have revised the manuscript to clarify the term 'annealing step' and provide more context for the reader. Specifically, we have updated the text on p. 12 to read: 'Specifically, an injection rate of 6 mL h⁻¹ was used for adding of 4.5 mL of InAs cluster solution (1.1 mg mL⁻¹) into 6.0 mL of InAs seed solution (1.3 mg mL⁻¹), attributing to one injection step, followed by a 30-minute annealing step at 300 °C. This process was repeated up to 16 times (8 injection and 8 annealing steps, or 8 growth cycles) and could be quenched after each step depending on the desired QD size.'

We also have revised the mentioned text passage as follows (p. 14): 'To access the role of annealing steps in our seeded growth method, we conducted a control experiment where cluster solution was continuously injected without any annealing steps in between (Supplementary Table 6). Also here, non-agglomerated large InAs QDs were produced, but the absorption edge reached only 1,750 nm after injection of the same amount (22.5 mL) of cluster precursor solution, compared to ~1,940 nm for the standard seeded growth method. The smaller QD size in the control experiment suggests that the annealing steps were crucial for promoting further QD growth, likely due to the lower reactivity of the cluster precursors.'

Finally, we've also updated the Methods section to clarify the annealing procedure: 'As soon as 300 °C were reached, 4.5 mL of InAs cluster solution was continuously injected into the seed solution

using a syringe pump at an injection rate of 6 mL h⁻¹. A 30-min annealing step at the same temperature followed. The injection-annealing cycles could be repeated multiple times, depending on desired QD size (see Supplementary Table 5). A control experiment was performed without the annealing steps (Supplementary Table 6).'

Once again, we sincerely thank the Reviewer for their valuable feedback and careful review of our manuscript.

Response to the Reviewer #2

We sincerely thank the Reviewer for their insightful suggestions and valuable feedback. We have carefully addressed the points raised in the review and made the following revisions.

This paper describes a novel method for growing large InAs QDs up to around 40 nm with high crystallinity and good size control starting from non-pyrophoric precursors. It addresses the long-standing challenge of extending the absorption range of InAs QDs in SWIR from 1600 nm to much longer wavelengths, beyond 2600 nm. The main approach relies on the use of well-defined InAs clusters obtained at lower temperature (180°C), which are either grown to 4 nm nanocrystals used in a seeded-growth reaction, or directly applied in a heat-up approach. The samples were characterized using standard techniques, in particular absorption spectroscopy, X-ray diffraction and TEM, and the main reaction parameters were explored to optimize the synthesis methods.

In the introduction and conclusion, a bigger accent could be put on the impact for SWIR applications, since it presents a core achievement of the work and there are not much other Pb- and Hg-free QDs that can reach this region.

We thank the reviewer for this insightful suggestion. We agree that emphasizing the potential of our InAs QDs for SWIR applications is crucial, especially in light of the limited availability of Pb- and Hg-free alternatives in this spectral range. We have accordingly revised both the introduction and the conclusion to better underscore the significance of these findings.

In the introduction, we added the following sections to highlight the relevance of SWIR applications:

- ‘Despite the availability of low-cost silicon-based detectors, their spectral sensitivity is limited to below ~1,100 nm.^[4,5] To access longer IR wavelengths, current technologies commonly rely on epitaxially grown materials such as indium gallium arsenide (InGaAs), which are costly due to energy-intensive deposition processes.^[5,6]’
- ‘Achieving larger sizes is essential for accessing deeper IR wavelengths and approaching bulk-like properties, but requires [...].’
- ‘The strategy also offers [...], paving the way for industrial production of RoHS-compliant, IR-active QDs. Our findings offer a new platform for fundamental studies and next-generation IR optoelectronic technologies, especially in the SWIR region.^[41,42]’

In the conclusion, we highlighted the potential for large-scale applications with the following addition: ‘The strategy has the potential for adaptation to continuous flow reactors, moving the field closer to large-scale production of RoHS-compliant Pb- and Hg-free NIR and SWIR CQDs. By unlocking

a new size regime and simplifying the synthetic toolbox for InAs NPs, this work lays a foundation for advances in IR optoelectronics, including biophotonics, telecommunications, and sensing technologies.'

All in all, this manuscript appears perfectly suitable for publication in Nature Communications, due to the originality of the approach and quality of the results, which open up new horizons.

We sincerely thank the reviewer for the positive evaluation and encouraging remarks regarding the originality and significance of our work.

Nonetheless there is a list of points which should be addressed:

1) The difference between seeds and clusters is a bit hard to follow, especially in the second part of the manuscript, I would suggest to elaborate in more details figure 1 with some key features of both. Fig. 1 could also be improved to make it visually more attractive.

We thank the reviewer for this valuable suggestion.

To improve clarity regarding the distinction between clusters and seeds, we have revised both the main text and Fig. 1. To avoid overcrowding the figure, we chose not to include detailed feature lists directly in Fig. 1, but instead clarified these points in the main text.

In the 'Synthesis of InAs clusters' section, we expanded our description of the cluster precursors (p. 4) and incorporated additional data into Fig. 2:

- 'For cluster synthesis, amino-As solution in oleylamine (OAm) was hot-injected into a mixture of InCl in OAm and trioctylphosphine (TOP) at 130 °C, and the reaction was maintained for 76 min. [...] In the high-resolution transmission electron microscopy (HR-TEM) images (Fig. 2b-d), small nanostructures can be seen. Some of the structures seem to form elongated aggregates, possibly during deposition process on the grids. Due to the low contrast, neither the crystallinity nor the shape of the clusters can be directly determined from the HR-TEM images.^{[41,42]'}
- '[...] synthesis of small InAs clusters (Fig. 1a), which exhibited optical activity in the VIS range'.

In the 'Synthesis of InAs seeds' section, we added (p. 6): 'The seeds showed lower size dispersity [...], and the corresponding XRD and HR-TEM data (Supplementary Fig. 10 and Supplementary Fig. 11) confirmed the crystalline nature of these particles.'

We believe that the updated Fig. 1 provides a clearer overview of our synthetic strategy, while the revised text more explicitly outlines the distinguishing features of clusters and seeds.

2) What are the reaction yields for the cluster synthesis and for the seeded-growth synthesis? insoluble side-products such as metallic indium (p. 3) were observed. In other words, how much, approximately, of the initially used InCl transforms into clusters and how much into In(0)? Does the yield change with bigger particles? Or between seedless and seeded growth?

We thank the reviewer for requesting a more quantitative description of the reaction yields.

In response, we have added information about the reaction yield and corresponding calculations to the revised manuscript: the cluster synthesis (Supplementary Note 3), the heat-up seed synthesis (Supplementary Note 4), the seeded growth approach (Supplementary Note 5), and the seedless growth of different sized QDs (Supplementary Note 6).

We also clarify that, prior to using clusters or seeds for either the seeded or seedless growth procedures, insoluble byproducts were always removed by centrifugation. Importantly, no insoluble byproducts were observed in either the seeded or seedless growth approaches, as confirmed by XRD and TEM analyses (Fig. 5–7 and Supplementary Fig. 27). To make this point clearer, we revised the main text (p. 7) as follows:

- ‘In the last stage of growing large InAs QDs *via* the seeded growth approach, we used InAs cluster and seed solutions after removing insoluble byproducts’.

Below we summarize the reaction yields presented in Supplementary Notes 4–6 and in the Methods section:

Cluster synthesis (relative to As): 49%

Cluster synthesis (relative to In): 16%

- InCl was used as a 3.1-fold excess, following Bawendi et al. (2020)

- Insoluble byproducts are therefore approx. 84% relative to InCl. explaining the pronounced brown-grey coloration before purification

- XRD (Supplementary Fig. 2) indicates these byproducts consist primarily of metallic In, with minor amounts of In₂O₃ and InAs

Seed synthesis (relative to As): 90%

Seed synthesis (relative to In): 23%

Seeded growth synthesis (16 steps): 76%

Seedless growth (10 steps): 98%

Seedless growth (6 steps): 114%

- Value slightly above 100% probably originated from the very small amount of inorganic material used for analysis (0.09%), which increased weighing uncertainty

As requested, we also compared the yields between seeded and seedless growth, and between different particle sizes (Supplementary Note 6):

- ‘By comparing the mass of the resulting InAs QD samples obtained at different synthesis stages of the seedless growth synthesis (15 mg after 6 steps, 25 mg after 10 steps), it is evident that continued QD growth takes place, which is consistent with observations from TEM analysis and absorption measurements. Since the seedless growth synthesis begins without any InAs material in the flask, the overall reaction yield after the same number of steps is lower than that observed in the seeded growth process.’

We hope these additions fully address the reviewer’s request for quantitative insight into reaction yields and byproduct formation.

3) Did the authors also try the seedless growth without annealing steps?

We thank the reviewer for this important question. Yes, we performed a control seedless growth experiment in which all annealing steps were omitted. This experiment is described on p. 13 of the revised manuscript (p. 9 in the original version):

- ‘In a control experiment, no annealing steps were conducted. Consistent with the control seeded growth experiment, non-aggregated, large InAs QDs were still produced. After injecting the same volume of cluster solution (22.5 mL) as in the standard seedless growth approach, the resulting QDs were 27 nm in size, compared to 40 nm when annealing steps were included (after precipitation techniques were applied). This suggests that alternating annealing steps play a crucial role in the growth process when using clusters as precursors.^{[41,42]’}

For completeness, we have now also added a brief description of this control experiment to the revised Methods section: ‘In a control experiment, no annealing steps were involved.’

4) In Fig.2 the T₀ spectrum could be added.

We thank the reviewer for this helpful suggestion. As requested, the T₀ spectrum has now been added to Fig. 2 (Fig. 3c in the revised version).

5) The absorption spectra of the >20 nm nanocrystals could have been taken on solid samples by applying diffuse reflectance spectroscopy instead of liquid phase measurements in TCE. Given the unique size regime of the obtained samples, this information would have been quite interesting and would potentially allow to draw figure S11 (which I suggest to put into the main manuscript) up to 40 nm. Also, it would allow to give a more precise evaluation of the absorption range covered by the obtained samples, the value of 2600 nm seems to be a rough estimation.

We thank the reviewer for this constructive suggestion and agree that solid-state spectroscopy data for the very large InAs QDs (< 20 nm) would significantly enhance the analysis. As recommended, we have now moved the plot from Fig. S11 to the main manuscript (Fig. 5b).

While diffuse reflectance spectroscopy (DRS) would be ideal for such measurements, we faced limitations due to unavailable instrumentation. Instead, we explored several alternative approaches for measuring solid samples.

Using a Cary 5000 spectrometer equipped with an integrating sphere, which is well-suited for scattering samples, we encountered limits above ~2,600 nm. Despite optimizing parameters such as integration time and slit width, no suitable baseline could be collected. We have mentioned this issue on p. 11 of the revised manuscript: 'Absorption measurements using an integrating sphere were not feasible in the MIR range due to technical limitations.'

As an alternative, we performed solid-state measurements on the Cary 5000 without an integrating sphere (same setup as for in-solution measurements). We observed a red-shifting absorption edge with increasing NP size. This setup allowed detection of absorption edges extending up to at least ~3,000 nm for the largest QDs. However, scattering effects persisted, reducing the quantitative reliability of the measurements and potentially obscuring the absorption features. For QDs larger than 28.5 nm (approaching the electron Bohr radius of InAs), absorption maxima are typically absent due to the weak confinement regime. These points are now discussed in the manuscript (p. 11):

- 'Therefore, additional measurements were conducted on a substrate (Supplementary Fig. 29b). However, scattering effects and ligand absorption features persisted above ~2,600 nm. Despite the absence of distinct absorption maxima, a redshift in the absorption edge was observed, suggesting quantum confinement effects.^[42]'
- 'In general, QDs sized between the electron Bohr radius (28.5 nm for InAs)^[51] and the exciton Bohr radius (30 – 44 nm for InAs)^[10,51,52] are expected to fall into the weak confinement regime, where absorption maxima are typically absent.^[51,53] This means in our case, that probably, no distinct absorption maxima would be observed for large InAs NPs in this size regime.^[42]'

To complement the solid-state measurements, we performed additional ATR-FTIR spectroscopy, which revealed size-dependent mid-IR absorption features (1,600 - 920 cm⁻¹), likely due to oleylamine (OAm) ligands. These results are shown in new Supplementary Fig. 29d-e. To aid in interpretation, we have included arrows indicating the estimated absorption maxima, calculated using the formula provided by Kuno et al. These data show a redshift

trend with increasing QD size, consistent with quantum confinement effects, and are discussed on p. 12:

- ‘Further ATR-IR analysis of InAs NP samples with average sizes ranging from 12 to 36 nm (Supplementary Fig. 29d-e) revealed, in addition to the ligand features, a size-dependent, broad absorption band in the mid-infrared (MIR) range (1,600 – 920 cm^{-1}), likely associated with the vibrational modes of OAm.^[42,56] As the QD size increased, this band shifted further into the red. Such a size-dependent effect is particularly interesting and likely arises from the significant size variation within the same material, which amplifies this effect.^{[42]’}

While ATR-IR measurements revealed a qualitative redshift with increasing QD size, no distinct absorption features were observed in the expected region. This is discussed on p. 12 and in Supplementary Fig. 29d-e:

- ‘The measurements revealed a qualitative redshift in absorption as the NP size increased. However, no distinct absorption features were observed, as expected, since ATR-IR characterizes the NP surface.^{[42]’}

We also address the ongoing debate surrounding the exciton Bohr radius of InAs (ranging from 30 – 44 nm). As noted on p. 12, based on our measurements, it remains unclear whether NPs larger than 30 nm can be classified as QDs. This is a fundamental question that warrants further study.

- ‘Given the ongoing debate surrounding the exciton Bohr radius of InAs QDs (ranging from 30 to 44 nm, as mentioned above)^[10,51,52], it remains uncertain whether InAs NPs larger than 30 nm can be classified as QDs. Even if they do, they likely would not exhibit distinct absorption bands, as they would fall into the weak confinement regime. Instead, they would be better classified as near-bulk QDs.^[51] Due to the addressed wavelength range, an extensive optical characterization of the large InAs particles’ properties requires further improvements. This characterization is within the scope of follow-up studies, where measurements on the near-bulk QDs could provide insights into the exciton Bohr radius of InAs QDs, a fundamental parameter in the study of QD systems. For the first time, such large colloidal InAs NPs are now experimentally accessible.^{[42]’}

6) In SI, Fig 1, not sure if the evolution is related to the reaction proceeding over longer times (two years!), or particles being slowly oxidized by traces of oxygen that might be present in the inert atmosphere.

We thank the reviewer for this insightful comment.

To clarify the nature of the spectral evolution, we compared the peaks from the 2-year sample with the absorption features of early-time aliquots from the cluster synthesis, as presented in Fig. 2 (now Fig. 3c). This comparison has been added to a new Supplementary Fig. 4.

The characteristic peaks at ~315 and ~350 nm are consistent and even more pronounced in the 2-year sample, indicating that the observed spectral changes are not due to slow oxidation.

Instead, they correspond to the formation of small VIS-active structures, similar to those observed during the early stages of cluster formation. This comparison supports the conclusion that the spectral evolution reflects chemical transformations linked to cluster formation rather than slow oxidation. We have included this discussion in the revised manuscript (p. 5):

- ‘We attribute the absorption features in the UV to molecular structures such as organometallic intermediates or ultra-small clusters. In a separate experiment (Supplementary Fig. 3), similar features were observed (Supplementary Fig. 4a). Comparing the corresponding absorption spectrum with those of early-stage aliquots taken during synthesis at 110 °C (see Fig. 3), we observed similar absorption maxima at ~315 and ~350 nm, with a higher intensity ratio (Supplementary Fig. 4b). This suggests that the absorption features correspond to the formation of specific small structures active in the VIS range.^{[41,42]’}

7) Experimental findings should not be discussed in the Conclusions but in the Discussion section. In particular, Fig. S16 (the RT evolution of clusters) is intriguing: the authors ascribe the sharp absorption features to the formation of metallic nanoparticles - are there any data supporting this hypothesis?

We thank the reviewer for this important suggestion. In response, we have revised the manuscript to move experimental findings into the Discussion section, as recommended. Specifically, Fig. S16 (now Supplementary Fig. 7) and the corresponding discussion have been relocated to the main manuscript (p. 5):

‘Interestingly, a room-temperature reaction proved unsuitable for producing InAs clusters (see Supplementary Note 1 and Supplementary Fig. 6). The absorption spectra of the particles formed were consistent with metallic nanostructures as presented in the in Supplementary Fig. 7 and Supplementary Fig. 8.^{[41,42]’}

Regarding the hypothesis about presence of metallic NPs, we conducted an experiment where we measured the absorption of NPs in ethanol (non-solvent) and compared it to their behavior in OAm (Supplementary Fig. 8). Upon redispersion in ethanol, the pink-purple NPs aggregated, which caused broadening of their absorption maxima and increased scattering. This spectral broadening upon aggregation is typical of metallic NPs due to localized surface plasmon resonance (LSPR), as reported previously for gold NPs by Raghavendra et al., 2013 [3]. These results support the hypothesis that the observed nanostructures are metallic NPs, not InAs clusters, and their characteristic pink-purple color arises from plasmonic resonance. We discuss these findings in Supplementary Note 1:

‘The absorption spectrum of the pink-red sample showed sharp peaks (Supplementary Fig. 7), suggesting the formation of metallic NPs. The plasmonic nature of these structures was further investigated in a separate experiment (Supplementary Fig. 8), where they exhibited aggregation, scattering, and broadening of absorption maxima in the non-solvent ethanol. This behavior is typical for metallic nanoparticles due to localized surface plasmon resonance (LSPR).^{[2,3]’}

8) The last phrase of the conclusions says that the continuous injection approach could be easily adapted to continuous flow synthesis. How could this be practically achieved? It seems that the continuous injection of precursors to a given reaction volume is indeed one step difficult to implement in continuous flow (?)

We thank the reviewer for this thoughtful question. It is correct that directly translating the continuous injection strategy into a continuous flow setup is challenging, as continuous injection into a fixed reaction volume does not directly apply to flow systems.

However, we believe that utilizing liquid In and As precursors, which are already present in the InAs cluster solution, represents a promising step toward their application in flow reactors. This approach also offers a safer and less toxic alternative to organometallic As precursors. One potential solution could involve introducing the cluster precursors at different points along the flow path or employing segmented reactors to better control the reaction process.

9) Given the size distribution, the specification of 40.5 nm looks like excessive precision, I would suggest throughout the manuscript to mention that the method gives access to QDs of up to 40 nm size.

We thank the reviewer for this helpful comment. In response, we have revised the manuscript to more generally refer to NPs of up to 40 nm in size.

10) Fig. 2: the inscription on the vials should be removed and eventually be replaced by a real legend.

We thank the reviewer for this suggestion. The handwritten inscriptions on the vials in Fig. 2 (now Fig. 3) have been removed and replaced by a legend.

11) A 1:1 In:As ratio has been applied for the cluster synthesis. In principle a 3:1 ratio would be required to reduce As from +3 to -3, it should be discussed why 1:1 is used.

We appreciate the reviewer's attention to stoichiometry. Upon careful review, we confirm that a 3:1 In:As ratio, consistent with the reduction of As^{3+} to As^{3-} , was indeed used in the cluster synthesis (as described in the Methods section).

We have now clarified this point also in the main text (p. 4): 'A 3.1:1 ratio of In:As as described in [26] was used.'

12) The use of the term "highly monodisperse" may not be appropriate for the clusters, as their absorption spectra are not that well defined showing a rather broad excitonic peak. For the TEM images a couple of size histograms would be helpful.

We thank the reviewer for this helpful comment. We agree that the term 'highly monodisperse' is not appropriate to clusters, given the broad absorption maximum (Fig. 2a) and the TEM images. To provide a more quantitative assessment, we have added a size histogram of the clusters used as precursors in Fig. 2b, along with HR-TEM images. Furthermore, we have removed the term "highly monodisperse" from the revised manuscript, as suggested.

13) The particles in Fig. 3a with an absorption maximum at 556 nm are estimated to have a size of 2.3 nm. Is this consistent with the sizing curve of InAs QDs? They could be likely smaller (cf Fig. S2)

We thank the reviewer for this valuable observation.

In the original version, the size was estimated on the long axis, which explains the larger value. However, after considering the short axis, which is more relevant for quantum confinement, we estimated the size to be 1.4 nm. This updated size has been reflected in Fig. 4a and Supplementary Fig. 9.

This value corresponds to a calculated excitonic peak at 461 nm. The discrepancy between the experimental absorption maximum (556 nm) and the calculated value (461 nm), which is approximately 100 nm, may arise from limitations in the formula used for size estimation. The formula was initially developed for QDs in the size range of 3.4 – 6 nm and may not fully account for the properties of smaller clusters. Additionally, determining the exact size and shape of the clusters from the microscopy images (Fig. 4a, Supplementary Fig. 9) remains challenging.

We appreciate the reviewer's observation, which has allowed us to update the cluster size estimate based on the short axis.

14) In the Introduction there is a lack of references and precision: use of vague phrases in the introduction (“extraordinary optoelectronic properties”, “crucial role”, “variety of materials”, “harsher reaction conditions”, “several disadvantages”). For example, while as stated no further developments of the In(I)Cl / aminopnictogen method have been reported for InAs QDs, this synthetic route has been used for InP and InSb, which could be mentioned in the introduction.

We thank the reviewer for this helpful comment.

The Introduction has been revised to use more precise language. Where such terms remain, they are now supported by specific examples and appropriate references:

- ‘Hence, harsher reaction conditions like higher temperatures are required, challenging the precise control of the size and the size distribution.’^[26-28]
- ‘[...] the initial thermolysis method had several disadvantages compared to TMSAs-based methods, including long reaction times of up to 6 days and the production of small, amorphous NPs with sizes up to 2 nm.’^[10,33]

Additionally, we have now cited relevant works in the introduction regarding the InCl/aminopnictogen approach applied to the synthesis of InP and InSb quantum dots:

- ‘This method was later adapted for the synthesis of InSb and InP QDs,^[36-38] but has not been further developed for InAs QDs.’

15) On p. 5 it is stated that keeping the insoluble side-products from the cluster synthesis improves the quality of the final QDs leading to larger sizes and narrower size distributions. Do the authors have any hypothesis on the mechanisms behind this behavior?

We thank the reviewer for this insightful question.

While the precise mechanism is not fully established, we hypothesize that the insoluble side-products from the cluster synthesis (primarily metallic indium) may act as a reservoir of indium species during the subsequent heat-up. This reservoir could provide a more controlled and steady supply of In precursors, promoting uniform growth of the QDs. Additionally, these byproducts might influence the local chemical environment, affecting nucleation and growth kinetics, which can lead to larger particle sizes and narrower size distributions.

We have added this discussion to the revised manuscript (p. 7): ‘TEM revealed the presence of larger, darker particles, likely metallic In (Supplementary Fig. 16), suggesting incomplete reaction in the initial cluster growth. These findings suggest that the byproducts may serve as a reservoir during the heat-up process, influencing the local chemical environment and promoting more uniform nucleation and growth of the QDs.’

16) Fig. S6: the reference diffraction pattern of InAs should be added. Idem Fig. S10

We thank the reviewer for the suggestion. Reference diffraction patterns of InAs have been added to both Fig. S6 (now Supplementary Fig. 10) and Fig. S10 (now Supplementary Fig. 20), as well as to all other XRD diffractograms presented in our work.

17) Fig. S7: why absorbance is given in arbitrary units?

We thank the reviewer for pointing this out. This was an oversight on our part — the absorbance in Fig. S7 (now Supplementary Fig. 12), as well as in other absorption spectra, should not have been labeled in arbitrary units. The data have now been updated with the correct absorbance scale in the revised version.

18) In the Manuscript and Supplementary Information "heating up" should be termed "heat-up" throughout the text.

We thank the reviewer for pointing out this terminology issue. All instances of 'heating up' have been revised to 'heat-up' throughout the manuscript and Supplementary Information.

Once again, we sincerely thank the Reviewer for their valuable feedback and careful review of our manuscript.

Response to the Reviewer #4

We sincerely thank the Reviewer for their insightful suggestions and valuable feedback. We have carefully addressed the points raised in the review and made the following revisions.

The authors reported the synthesis of **free-standing** InAs quantum dots by wet-chemistry. Their main goal was to synthesize ultra-large InAs quantum dots. In order to reach their goal, the chosen synthesis protocol is a wet-chemical synthesis one that involves three steps. The authors mainly used UV-visible-NIR spectroscopy, Transmission Electron Microscopy and X-ray diffraction to characterize their samples.

Here are my comments:

- Remarks on the text.

1. The authors claimed that they have InAs quantum dots but nowhere they experimentally proved that they had quantum dots and not just InAs nanoparticles. Authors could use ARPES, Raman spectroscopy or EELS spectroscopy to confirm that indeed they have quantum confinement on their InAs quantum dots.

We thank the reviewer for raising this important point. We fully agree that direct demonstration of quantum confinement is essential to distinguish true quantum dots from larger, bulk-like nanoparticles.

In our study, we observed a clear size-dependence in the optical absorption spectra of particles up to ~18 nm in diameter (Fig. 4a, corresponding to Fig. 5a in the revised manuscript). These spectra showed well-defined first excitonic transitions that redshifted with increasing NP size, in agreement with quantum confinement. Additionally, a comparison with the analytical expression for the first exciton transition peak energy for InAs QDs revealed good correspondence in this size range (Supplementary Fig. 11, now Fig. 5b in the revised version).

Particles larger than ~19 nm were colloiddally unstable, and measurements in solution (including those after shaking the sample) resulted in scattering effects and overlap with ligand absorption beyond ~2,600 nm. We pointed this out in the revised manuscript as well (p. 11): 'Absorption spectra of large, colloiddally unstable InAs NPs in TCE (Supplementary Fig. 29a) showed

absorption in the range up to $\sim 2,600$ nm, but without distinct absorption maxima. Beyond $\sim 2,600$ nm, scattering from NPs and overlap with ligand absorption interfered with the measurements.^[41,42]

We truly appreciate the suggested techniques and have carefully reviewed the possibility of characterizing our colloidal nanoparticles larger than 18-19 nm. While ARPES, Raman spectroscopy, and EELS are useful for epitaxial QDs, they are not standard characterization methods for colloidal InAs quantum dots. Their applicability to our system is limited due to the following reasons. Our particles are non-emissive, ruling out photoluminescence-based confirmation. ARPES requires single-crystal epitaxial thin films and ultra-high vacuum, which are incompatible with our colloidal systems. EELS, while potentially offering insights into the electronic structure, is not typically used to demonstrate quantum confinement in colloidal QDs capped with organic ligands. Raman spectroscopy is limited by weak signal intensity and overlapping modes, especially for colloidal QDs. Therefore, we have focused on well-established optical characterization techniques. Absorption measurements (including measurements on substrates) and ATR-IR spectroscopy were used to assess quantum confinement in our NPs.

To minimize scattering effects during absorption measurements, we attempted to use an integrating sphere; however, this approach was not feasible due to technical limitations (p. 11): ‘Absorption measurements using an integrating sphere were not feasible in the MIR range due to technical limitations.’

Substrate measurements, while still showing scattering effects, revealed a redshift trend in absorption, suggesting quantum confinement (p. 11): ‘Therefore, additional measurements were conducted on a substrate (Supplementary Fig. 29b). However, scattering effects and ligand absorption features persisted above $\sim 2,600$ nm. Despite the absence of distinct absorption maxima, a redshift in the absorption edge was observed, suggesting quantum confinement effects.’^[42]

Regarding larger NPs, we discussed the weak confinement regime as a possible explanation for the absence of distinct absorption maxima. For larger NPs, the quantum dots are likely to behave as near-bulk particles, leading to broader absorption features (p. 11): ‘In general, QDs sized between the electron Bohr radius (28.5 nm for InAs)^[51] and the exciton Bohr radius (30 – 44 nm for InAs)^[10,51,52] are expected to fall into the weak confinement regime, where absorption maxima are typically absent.’^[51,53] This means in our case, that probably, no distinct absorption maxima would be observed for large InAs NPs in this size regime.^[42]

Additionally, ATR-IR measurements revealed a qualitative redshift in absorption with increasing NP size, but no distinct absorption features were observed, as expected for this method (p. 12 and Supplementary Fig. 29d-e): ‘The measurements revealed a qualitative redshift in absorption as the NP size increased. However, no distinct absorption features were observed, as expected, since ATR-IR characterizes the NP surface.’^[42]

In addition, a size-dependent, broad absorption band in the MIR range between $1600-920\text{ cm}^{-1}$ (6,250 – 10,870 nm) was observed in the ATR-IR measurements, likely associated with oleylamine vibrational modes (p. 12 and Supplementary Fig. 29d-e): ‘Further

ATR-IR analysis of InAs NP samples with average sizes ranging from 12 to 36 nm (Supplementary Fig. 29d-e) revealed, in addition to the ligand features, a size-dependent, broad absorption band in the mid-infrared (MIR) range (1,600 – 920 cm^{-1}), likely associated with the vibrational modes of OAm.^[42,56] As the QD size increased, this band shifted further into the red. Such a size-dependent effect is particularly interesting and likely arises from the significant size variation within the same material, which amplifies this effect.^[42]

Lastly, we also addressed the ongoing debate regarding the exciton Bohr radius of InAs QDs (30 – 44 nm). As noted on p. 12, we cannot definitively classify NPs larger than 30 nm as QDs based on our current measurements. Whether these particles can be considered QDs remains an open question, and further studies are needed to explore this fundamental question: ‘Given the ongoing debate surrounding the exciton Bohr radius of InAs QDs (ranging from 30 to 44 nm, as mentioned above)^[10,51,52], it remains uncertain whether InAs NPs larger than 30 nm can be classified as QDs. Even if they do, they likely would not exhibit distinct absorption bands, as they would fall into the weak confinement regime. Instead, they would be better classified as near-bulk QDs.^[51] Due to the addressed wavelength range, an extensive optical characterization of the large InAs particles’ properties requires further improvements. This characterization is within the scope of follow-up studies, where measurements on the near-bulk QDs could provide insights into the exciton Bohr radius of InAs QDs, a fundamental parameter in the study of QD systems.’

As a result of these findings, we have revised the manuscript to describe InAs particles larger than 30 nm as nanoparticles, rather than quantum dots. We also updated the manuscript title to reflect this distinction.

We hope the revised version clarifies the concerns regarding the classification of larger InAs particles and provides a more accurate representation of our findings.

2. What is the energy bandgap of your InAs quantum dots?

We thank the reviewer for this question.

The energy bandgap of the InAs QDs in our study depends on their size, as a result of quantum confinement effects, and varies systematically with particle diameter. The energy bandgap was determined from the first excitonic absorption maxima observed in the UV–vis–NIR spectra.

For InAs QDs in the size range of ~6 – 18 nm, the excitonic peaks correspond to bandgaps ranging from ~1.1 eV to 0.54 eV (Fig. 5a-b in the revised manuscript). For larger NPs beyond 18 nm, we were unable to identify distinct absorption maxima, due to factors such as increased scattering, reduced colloidal stability, overlap with ligand absorption features in the SWIR, and detector limitations. As discussed in a previous response, we cannot definitively classify the 40 nm particles as QDs. Based on the absorption edge observed around 3,000 nm in Supplementary Fig. 29b, we roughly estimate the corresponding bandgap to be 0.4 eV.

3. How polydisperse is your population of InAs quantum dots? Could you perform Dynamic Light Scattering (DLS) measurements on your colloids?

We thank the reviewer for this important question.

While DLS is a common technique for analysis of colloids, it is not well suited for our system for several reasons. DLS measures the hydrodynamic diameter, which includes contributions from the organic ligands and solvation layers, not just the inorganic core. For larger QDs (above ~19 nm), colloidal stability is significantly reduced, leading to aggregation and strong scattering, which can compromise the reliability of DLS results. Furthermore, while DLS works well for larger colloids (>100 nm), its reliability decreases for particles <100 nm, especially in the presence of ligands and solvent shells, all of which strongly affect the measured hydrodynamic diameter.

To address this important question, we have provided TEM images and size distribution histograms of the samples without size selection in the revised manuscript (Supplementary Fig. 19, Supplementary Fig. 22, and Supplementary Fig. 23). As mentioned in the original manuscript, the growth of large NPs was not monodisperse, and precipitation techniques were required for NPs obtained from both seeded and seedless growth methods. To clarify the use of size-selective precipitation, we revised the manuscript text (p. 8): 'The samples were purified post-synthetically using size-selective precipitation techniques (hereafter referred to as 'precipitation techniques'), as described in the Methods section.' Additionally, we updated the description later in the manuscript (p. 8): 'Fig. 5 summarizes the data of the resulting InAs QDs after applying the precipitation techniques.'

We also provided a more detailed description of precipitation process in the Methods section: 'For the isolation of monodisperse InAs QD fractions, size-selective precipitation techniques were applied using anhydrous toluene and ethanol as the solvent and non-solvent, respectively. First, the reaction mixture was shaken and centrifuged to separate QDs with sizes above ~9 nm, which were colloidally unstable in OAm at r. t., but stable in toluene and TCE for sizes up to ~19 nm. If a pellet formed, it was dispersed in toluene or TCE, and centrifuged to separate QDs with sizes above ~19 nm. In the last step, if a pellet formed, it was dispersed in TCE or toluene, and ethanol was added dropwise until the mixture became turbid. The mixture was then centrifuged, and the pellet dispersed in TCE for characterization. If no pellet formed, the supernatant containing QDs was size-selective precipitated using ethanol as non-solvent, as described above. A rainbow-like shimmer in areas of thin pellet thickness indicated interference effects, suggesting a narrow size distribution of the NPs.^[42]'

4. The authors claim that they synthesize very large InAs quantum dots. So, what is the current state of the Art? What is the largest InAs quantum dot as of today? The problem is that I found a paper published by Saito et al. in Applied Physics Letters (APL) that already claimed in 1999 their ability of synthesizing very large InAs quantum dots with sizes larger than the ones you are currently reporting (~40 nm). Look specifically at the Figure 2 published in APL (1999), vol. 74, pages 1224-1226.

We thank the reviewer for pointing out the work by Saito et al. (1999). We are aware of this important contribution and would like to clarify a key distinction: the InAs quantum dots reported in that paper were grown epitaxially on GaAs substrates using molecular beam epitaxy (MBE). These are not colloidal quantum dots, and their growth mechanism differs fundamentally from the colloidal synthesis methods reported in our study.

The key difference is that MBE-grown InAs quantum dots form *via* an entirely different mechanism, typically resulting in island-like structures that are confined within the substrate, whereas our work deals with solution-processed, colloidal InAs quantum dots that are free-standing and synthesized in a liquid phase. The synthesis methods, structural characteristics, and the resulting properties of these two types of quantum dots are fundamentally different. The largest colloidal InAs QDs reported to date are ~13 nm (p. 3): For example, this was demonstrated when the largest non-elongated InAs QDs reported to date were produced by this method, achieving an absorption peak in the range up to ~1,850 nm and QD sizes up to ~13 nm.^[34]

To clarify this distinction, we have added a reference to the work of Saito et al. in the revised manuscript (Introduction, p. 2): 'Although large InAs QDs grown epitaxially have been reported 1999,^[29] the colloidal synthesis of QDs larger than 7 nm, with excitonic absorption peak above 1,400 nm, remains a significant challenge.^[11,24,30]'

This addition should help make the distinction between epitaxially grown and colloidal synthesized InAs quantum dots clearer to the reader.

5. The bibliography contains only 22 references which is very low for a topic like this one. A lot of work has been published on quantum dots and specifically on InAs quantum dots. According to the author guidelines of Nature Communications, you are allowed to reference 70 articles. So, you have a lot of room to improve the bibliography and context of your article.

We thank the reviewer for the valuable suggestion regarding the expansion of the bibliography. In response, we have significantly extended the list of references to 56 in the revised manuscript to better reflect the broad context of research.

6. The acronym RoHS is used in the abstract (line 9). For clarity, acronyms should be avoided in abstract and RoHS should be spelled out completely.

We thank the reviewer for this helpful comment. We have now replaced 'RoHS' with 'Restriction of Hazardous Substances' in the abstract for clarity.

Remarks on the figures.

1. The sketch used to describe the three steps synthesis protocol in Figure 1 could be improved to be more visual than just using rectangles.

We thank the reviewer for the helpful suggestion. The schematic in Fig. 1 has been updated with more visual graphics. We believe this will improve clarity and reader understanding.

2. On the y-axis of Figure 2, absorbance does not have any units; not even arbitrary units because absorbance is by definition a logarithm and as you probably know a logarithm does not have any units.

We thank the reviewer for pointing this out. Absorbance is indeed a logarithmic quantity and, as such, has no units. In the revised manuscript and supplementary materials, we have updated the y-axis label accordingly, including for Fig. 2 (now Fig. 3c in the revised version).

3. The previous remark also applies to Figure 3 d, Figure 3e and Figure 4a.

We thank the reviewer for the additional note. The y-axis labels in Figures 3d (now Fig. 4d), 3e (now Fig. 4g), and 4a (now Fig. 5a) have been updated accordingly.

4. On Figure 3f, the error bar and the announced value should have the same rank! If you have an error of 1.2 nm then the announced value should be 3.5 nm.

We thank the reviewer for this valuable observation. The reported values in Fig. 3f (now Fig. 4e), as well as in Figs. 4a-c and 4f, have been corrected to align with the precision of the error bars.

5. The error bars on Figures 4b to 4e are wrong.

We thank the reviewer for the additional note. The error bars in Figures 4b-e (now Fig. 5c-f) have been corrected accordingly.

6. The error bars on Figures 5a and 5b are wrong.

We thank the reviewer for this observation. The error bars in Figures 5a and 5b (now Fig. 6a-b) have been updated accordingly.

7. On Figures 5a and 5b, could you rotate the figures or just the scale bar in order to have the scale bar horizontal? It will be easier for the readers to have all scale bars horizontal.

We thank the reviewer for this helpful suggestion. We have rotated the images in Figures 5a and 5b (now Fig. 6a-b) so that the scale bars are now horizontal, enhancing readability.

8. On Figure 5c, if you normalize the intensity, then the values on the y-axis go from 0 to 1 and should be displayed. If you want you could display your y-axis from 0% to 100%.

We thank the reviewer for the suggestion. We have updated Fig. 5c (now Fig. 6c) to include the reference for cubic InAs for better clarity. We chose not to display the y-axis values from 0 to 1 in order to maintain consistency with the style used across all figures in the manuscript.

9. The error bars on Figures 6a to 6d are wrong.

We thank the reviewer for the additional note. We have updated the error bars in Fig. 6a-d (now Fig. 7a-d) accordingly.

10. On Figures 6a, 6b, 6c and 6d, could you rotate the figures or just the scale bar in order to have the scale bar horizontal? It will be easier for the readers to have all scale bars horizontal.

We thank the reviewer for this helpful suggestion. We have rotated Figures 6ad (now Fig. 7a-d) so that all scale bars are now displayed horizontally, ensuring consistency across the figures and improving readability for the reader.

Once again, we sincerely thank the Reviewer for their valuable feedback and careful review of our manuscript.

Response to the Reviewer #1

I thank the authors for addressing all the reviewer's comments/concerns regarding this manuscript, it has greatly improved.

I believe the manuscript will be of great interest to readership of Nature Communication and the community working on colloidal InAs Quantum dots.

In my view, the revised manuscript can be accepted for publication.

We sincerely thank the Reviewer for their positive feedback.

We also thank them once again for their insightful suggestions that helped improve our manuscript.

Response to the Reviewer #2

The authors have carefully addressed all points I raised, and I recommend the publication of this article in its present form. Once again I would like to point out the high importance of the presented results for this field.

We sincerely thank the Reviewer for their positive feedback.

We also thank them once again for their insightful suggestions that helped improve our manuscript.

Maybe one small final comment concerning the reaction yield calculations: the molar quantity given (0.316 mmol, cf. Suppl. note 3) is not that of InAs clusters but that of InAs units and it does not appear to indicate the reaction yield according to In: due to the reaction mechanism, In has to be used with a 3:1 excess compared to As. Therefore, just giving the reaction yield according to As will be good enough.

We sincerely thank the reviewer for this helpful comment. We have chosen to report the reaction yield according to In as well, to highlight the high amount of unreacted In precursor. A similar calculation is also shown in the case of InAs seed synthesis (Suppl. Note 4).

Response to the Reviewer #4

The authors have carefully addressed the comments from all reviewers. The clarity of the text has been improved.

We sincerely thank the Reviewer for their positive feedback.

We also thank them once again for their insightful suggestions that helped improve our manuscript.

Manuscript #: NCOMMS-25-25803-T

Manuscript Title: Colloidal Synthesis of Ultra-Large InAs Quantum Dots 1

Authors: Ekaterina Salikhova, Alf Mews, Jan Steffen Niehaus

The authors reported the synthesis of free-standing InAs quantum dots by wet-chemistry. Their main goal was to synthesize ultra-large InAs quantum dots. In order to reach their goal, the chosen synthesis protocol is a wet-chemical synthesis one that involves three steps. The authors mainly used UV-visible-NIR spectroscopy, Transmission Electron Microscopy and X-ray diffraction to characterize their samples.

Here are my comments:

- Remarks on the text.
 1. The authors claimed that they have InAs quantum dots but nowhere they experimentally proved that they had quantum dots and not just InAs nanoparticles. Authors could use ARPES, Raman spectroscopy or EELS spectroscopy to confirm that indeed they have quantum confinement on their InAs quantum dots.
 2. What is the energy bandgap of your InAs quantum dots?
 3. How polydisperse is your population of InAs quantum dots? Could you perform Dynamic Light Scattering (DLS) measurements on your colloids?
 4. The authors claim that they synthesize very large InAs quantum dots. So, what is the current state of the Art? What is the largest InAs quantum dot as of today? The problem is that I found a paper published by Saito *et al.* in Applied Physics Letters (APL) that already claimed in 1999 their ability of synthesizing very large InAs quantum dots with sizes larger than the ones you are currently reporting (~40 nm). Look specifically at the Figure 2 published in APL (1999), vol. 74, pages 1224-1226.
 5. The bibliography contains only 22 references which is very low for a topic like this one. A lot of work has been published on quantum dots and specifically on InAs quantum dots. According to the author guidelines of Nature Communications, you are allowed to reference 70 articles. So, you have a lot of room to improve the bibliography and context of your article.
 6. The acronym RoHS is used in the abstract (line 9). For clarity, acronyms should be avoided in abstract and RoHS should be spelled out completely.

- Remarks on the figures.

1. The sketch used to describe the three steps synthesis protocol in Figure 1 could be improved to be more visual than just using rectangles.
2. On the y-axis of Figure 2, absorbance does not have any units; not even arbitrary units because absorbance is by definition a logarithm and as you probably know a logarithm does not have any units.
3. The previous remark also applies to Figure 3 d, Figure 3e and Figure 4a.
4. On Figure 3f, the error bar and the announced value should have the same rank! If you have an error of 1.2 nm than the announced value should be 3.5 nm.
5. The error bars on Figures 4b to 4e are wrong.
6. The error bars on Figures 5a and 5b are wrong.
7. On Figures 5a and 5b, could you rotate the figures or just the scale bar in order to have the scale bar horizontal? It will be easier for the readers to have all scale bars horizontal.
8. On Figure 5c, if you normalize the intensity, then the values on the y-axis go from 0 to 1 and should be displayed. If you want you could display your y-axis from 0% to 100%.
9. The error bars on Figures 6a to 6d are wrong.
10. On Figures 6a, 6b, 6c and 6d, could you rotate the figures or just the scale bar in order to have the scale bar horizontal? It will be easier for the readers to have all scale bars horizontal.